# Exploring channel distinguishability in local neighborhoods of the model space in quantum neural networks

**Sabrina Herbst**[1][*]**Sandeep Suresh Cranganore**[1,2]**, Vincenzo De Maio**[1][†]**& Ivona Brandić**[1]
[1] HPC Research Group, Faculty of Informatics, TU Wien, Vienna, Austria
[2] Peter Grünberg Institute-8, Forschungszentrum Jülich GmbH, Juelich, Germany

## Abstract

With the increasing interest in Quantum Machine Learning, Quantum Neural Networks (QNNs) have emerged and gained significant attention. These models have, however, been shown to be notoriously difficult to train, which we hypothesize is partially due to the architectures, called ansatzes, that are hardly studied at this point. Therefore, in this paper, we take a step back and analyze ansatzes. We initially consider their expressivity, i.e., the space of operations they are able to express, and show that the closeness to being a 2-design, the primarily used measure, fails at capturing this property. Hence, we look for alternative ways to characterize ansatzes, unrelated to expressivity, by considering the local neighborhood of the model space, in particular, analyzing model distinguishability upon small perturbation of parameters. We derive an upper bound on their distinguishability, showcasing that QNNs using the Hardware Efficient Ansatz with few parameters are hardly discriminable upon update. Our numerical experiments support our bounds and further indicate that there is a significant degree of variability, which stresses the need for warm-starting or clever initialization. Altogether, our work provides an ansatz-centric perspective on training dynamics and difficulties in QNNs, ultimately suggesting that iterative training of small quantum models may not be effective, which contrasts their initial motivation.

## 1 Introduction

With the increasing computational requirements of state-of-the-art ML models, the limitations of classical hardware, as theorized by Moore's law (Shalf, 2020), are becoming increasingly prevalent (Schulz et al., 2021; Reed et al., 2022). Therefore, alternative computing paradigms are heavily studied and the immense potential of Quantum Computing, combined with recent advances in quantum hardware, have made Quantum Machine Learning (QML) a promising candidate.

Within that framework, Quantum Neural Networks (QNNs) are heavily studied, as they are suitable for near-term hardware. Besides initial successes, however, the *Barren Plateau* phenomenon has been proven under various circumstances, meaning that gradients vanish exponentially in the number of qubits, leading to essentially flat loss landscapes that affect trainability (Cerezo et al., 2021a).

In light of these issues, we hypothesize that the architectures, called *ansatzes*, are part of the problem. QNN ansatzes consist of fixed and trainable gates, however, due to hardware limitations, currently mainly so-called Hardware Efficient Ansatzes (HEAs) (Kandala et al., 2017) are used, meaning they are built primarily based on hardware constraints. Therefore, we hypothesize that executing the circuit does not lead to a meaningful feature representation.

Currently, it is unclear how individual operations affect the model space, or, when including data, the data representation. There are no standardized ansatzes for Machine Learning (ML) and, due to the many degrees of freedom, ansatz tuning poses an optimization problem with intractable search

---
[*]sabrina.herbst@tuwien.ac.at
[†]vincenzo@ec.tuwien.ac.at

space. Further, there is little research on the ansatzes themselves, i.e., independently of data and loss, and it is non-obvious how feature extraction works in a QNN.

In particular, in classical ML, a vanilla CNN is ultimately designed to consider local features in lower layers, before considering global features in deeper layers, independently of the dataset. Transformers (Vaswani et al., 2017) work similarly, however, the trainable one-qubit and fixed two-qubit gates that are used in QNNs do not allow drawing intuitive conclusions about how features are extracted.

Therefore, the goal of this paper is to analyze and characterize QNN architectures. We aim to take a first step at exploring ansatzes and the model space, initially by considering the expressivity of ansatzes. Using tools from Quantum Information Theory and extensive numerics, we later analyze how the quantum circuit, i.e., the applied operation, changes upon parameter update, thus, providing insights into the model space beyond ansatz expressivity. Our contributions are as follows.

- Based on literature, we argue that *closeness to a 2-design*, a widely used measure for ansatz expressivity, does not adequately represent the possible set of unitary operations.

- To investigate the model space, we study how an ansatz changes upon parameter update for different HEAs. In particular, we introduce a measure for the distinguishability of an ansatz upon parameter update and provide an upper bound. It shows that many parameters are required to be able to distinguish the two operations, contrasting with the initial framework.

- We provide evidence that HEAs are hardly distinguishable upon parameter update in (1), random perturbations and (2), during training, even in early stages with larger updates.

- While not strictly equal, the observed behavior and effects thereof have remarkable similarities to Barren Plateaus, insinuating two perspectives on similar trainability issues. The results suggest that the architectures, independently of data and loss, may be flawed and play a fundamental role in the trainability problems observed today.

The paper is structured as follows. In Section 2, we provide background information on QNNs, existing expressivity measures and related work[1]. In Section 3, we discuss the closeness to being a 2-design as a measure of expressivity and Section 4 introduces a measure of distinguishability of quantum channels. We elaborate on our experimental setup in Section 5, and present our results in Section 6. Section 7 discusses implications of our results and Section 8 concludes the paper.

## 2 BACKGROUND

### 2.1 QUANTUM NEURAL NETWORKS

The small systems and the noise in today's Noisy Intermediate-Scale Quantum (NISQ) technology (Preskill, 2018), prevents execution of large-scale circuits. Consequently, so-called Variational Quantum Circuits (VQC) are applied, which are hybrid models with classical and quantum parts.

Simply put, a QNN consists of (1) an input state $|\psi_{\boldsymbol{X}}\rangle$, (2) an ansatz $U(\boldsymbol{\vartheta})$, (3) a loss function, and (4) an optimizer. The features are encoded onto the quantum state by means of a unitary feature encoding method $V(\boldsymbol{X})$, and the initial state is obtained as $|\psi_{\boldsymbol{X}}\rangle = V(\boldsymbol{X})|0\rangle^{\otimes n}$, with $n$ as the number of qubits.

HEAs consists of $L$ layers of trainable one-qubit rotation gates $V_i(\vartheta_i)$ and fixed two-qubit entangling gates $W_i$, with $i \in [1, L]$, expressed as Equation 1. A measurement is taken in the end, which is done through a Hamiltonian observable $\hat{O}$, and the result is expressed as the expectation value of the ansatz with respect to $|\psi_{\boldsymbol{X}}\rangle$ and $\hat{O}$, as shown in Equation 2.

$$U(\boldsymbol{\vartheta}) = \prod_{i=1}^{L} V_i(\vartheta_i)W_i = V_L(\vartheta_L)W_L \ldots V_1(\vartheta_1)W_1 \tag{1}$$

$$f(\boldsymbol{X}, \boldsymbol{\vartheta}) = \langle\psi_{\boldsymbol{X}}| U^{\dagger}(\boldsymbol{\vartheta})\hat{O}U(\boldsymbol{\vartheta}) |\psi_{\boldsymbol{X}}\rangle \tag{2}$$

---

[1]We refer to Appendix A.1 for a brief overview of Quantum Computing.

The loss is calculated with a classical loss function (e.g., the mean-squared error) on the predicted value (the expectation value of the QNN) and the target value, and passed to the classical optimizer. The optimizer computes the parameter updates and the circuit is again executed on the quantum computer, until some stopping criterion is fulfilled. As the optimization is done classically, this approach has the advantage of allowing a shallow quantum circuit, which limits the propagation of errors during execution in light of the significant amount of noise in today's hardware.

While HEAs are shallow, they are highly expressive (Holmes et al., 2022), which is a main promise of QML stemming from the significantly larger Hilbert space. Recent works, however, uncovered that loss landscapes of today's QNNs are plagued by Barren Plateaus (BPs), meaning exponentially vanishing gradients in problem size, leading to untrainable models (McClean et al., 2018).

## 2.2 ANSATZ EXPRESSIVITY

We acknowledge a missing clear definition of expressivity in QML for now. It is sensible, therefore, to transfer it from classical ML, where model expressivity is the range of functions a model can compute (Raghu et al., 2017). Hence, it is a function of architecture $A$, input $x$ and parameters $\vartheta$; $F_A(x; \vartheta)$. On the contrary, our work lies in data-agnostic characterization of models. To this end, we will only consider *ansatz expressivity*; the set of functions generated solely by an ansatz itself.

The most commonly used method to evaluate ansatz expressivity is to compare the probability distribution of the unitaries of the ansatz to the *Haar measure*[2], the uniform distribution over all unitaries $\mathcal{U}(d)$ (Cerezo et al., 2021a). If the probability distribution of the ansatz follows the Haar measure up to the $t$-th moment, it is considered a *t-design*. Their construction requires exponential time (Harrow & Low, 2009), however, approximate t-designs can be built efficiently (Dankert et al., 2009).

Closeness to a 2-design as a measure of expressivity for ansatzes was first defined for the $|0\rangle$ input state by Sim et al. (2019). In Holmes et al. (2022), it is expanded to a specific input state (i.e., with encoded data) and the measurement operator, as shown in Equation 3. There, an ansatz is viewed as an ensemble of unitary transformations $\mathbb{U} = \{U(\vartheta), \forall \vartheta \in \Theta\}$ over all possible parameters $\Theta$.

$$A_{\mathbb{U}}^{(t)}(\cdot) = \int_{\mathcal{U}(d)} d\mu(V) V^{\otimes t}(\cdot)(V^\dagger)^{\otimes t} - \int_{\mathbb{U}} dU U^{\otimes t}(\cdot)(U^\dagger)^{\otimes t} \quad \forall\, V \in \mathcal{U}(d) \tag{3}$$

## 2.3 RELATED WORK

BPs have been proven for deep (McClean et al., 2018) and shallow circuits with global measurement (Cerezo et al., 2021b), expressivity (Holmes et al., 2022; McClean et al., 2018), noise (Wang et al., 2021), entanglement (Ortiz Marrero et al., 2021) and it has been linked to cost concentration and narrow gorges (Arrasmith et al., 2022). Further, even shallow models without BPs have only a small fraction of local minima close to the global minimum (Anschuetz & Kiani, 2022)[3].

The problem has been linked to the curse of dimensionality in Cerezo et al. (2024), therefore, limiting the accessible Hilbert space was proposed. However, they showed that such models can be simulated classically, given an initial data acquisition phase on a quantum computer. This was extended to a trade-off between trainability and dequantizability in Gil-Fuster et al. (2024).

Besides that, parameter initialization strategies have been proposed, with some even providing better bounds on gradient magnitudes (Rad et al., 2022; Wang et al., 2023; Kulshrestha & Safro, 2022; Grant et al., 2019), however, the results of Herbst et al. (2024) imply that it is not the type of statistical distribution, but rather the used parameter ranges that could lead to better starting points.

Further, expressivity is commonly discussed to prove a quantum advantage with respect to classical models. The closeness to being a 2-design is the most widely employed measure, however, others have been proposed. In particular, Equation 3 is related to the Frame Potential (Gross et al., 2007; Roberts & Yoshida, 2017; Nakaji & Yamamoto, 2021). Another approach is to analyze the entangling capability as in Sim et al. (2019), i.e., the entanglement of the produced states. Other measures are the Covering Number from Du et al. (2022), analyzing Fourier coefficients (Schuld et al., 2021;

---

[2]We refer to Appendix A.3 for a short introduction to unitary groups and the Haar measure.
[3]For further information, we refer to Larocca et al. (2024), who published an excellent review on the topic.

Caro et al., 2021), or the Effective Dimension from Abbas et al. (2021). Further, one can consider *unitary t-designs*, i.e., consider the operations without the input state[4]. We employ the closeness to being a 2-design in our study, as it is frequently used, due to its intuitive interpretation. Beyond that, due to computational complexity, there is little research on properties of the architectures for now.

Moreover, the field of QML has seen a rise in works on efficient design of ansatzes. Recent works, such as Leone et al. (2024), have actively highlighted that well-defined ansatzes are a crucial step for large-scale deployment. In the field of ML, such approaches particularly include the field of Geometric Quantum Machine Learning (GQML) (Ragone et al., 2023; Larocca et al., 2022; Tüysüz et al., 2024; Wiersema et al., 2024), or adaptive ansatzes (Bilkis et al., 2023).

Further, the practical utility of the HEA has already been questioned and studied by Leone et al. (2024). They find that problems with product input states and data that satisfies a volume law of entanglement should avoid using such ansatzes, whereas area law of entanglement data leads to optimization landscapes without BPs. Moreover, the necessity of understanding and characterizing ansatzes has been discussed in Zhang et al. (2024), as well as in Larocca et al. (2023).

Changes in quantum channel or loss landscape upon perturbing parameters has not yet been studied for QNNs. However, perturbation analysis and sensitivity analysis have emerged as tools in classical ML. Examples are perturbing the input for explainability (Ivanovs et al., 2021; Pizarroso et al., 2022; Fel et al., 2023), or checking for stability (Testa et al., 2024). Similarly, weights can be perturbed for trainability (Wen et al., 2018), or to increase robustness and generalization (Wu et al., 2020; He et al., 2019; Dumford & Scheirer, 2020). To the best of our knowledge, there is no research on perturbing parameters to check how the underlying function changes, as, due to the stochasticity of quantum computing, this research question seems to be much more relevant in the quantum realm.

## 3   2-DESIGN AS A MEASURE OF EXPRESSIVITY

While 2-designs have turned out to be sufficient for many quantum information processing protocols (Holmes et al., 2022; Harrow & Low, 2009), in this section, we summarize and connect results from the literature to argue that closeness to a 2-design is an inadequate measure for model expressivity (i.e., the ability to sufficiently approximate the unitary group to a high degree).

For the argument we will use the so-called *Welch Bounds* which, loosely put, are inequalities related to evenly distributing a finite number of unit vectors in a vector space. This inequality was addressed in Welch (1974), where a lower bound on the inner-product between unit vectors was provided. Intuitively, a smaller lower bound corresponds to more evenly spread vectors in the vector space, i.e., a more uniform distribution.

Quantum **state t-designs** reduce integrals of polynomials over all quantum states to averages over a discrete set (cf. Equation 19). These are probability distributions over pure quantum states that replicate properties of the uniform (Haar) measure on the quantum states up to the $t$-th moment. We measure expressivity of 2-design based architectures by comparing (generalized) Welch bound (Scott, 2008) values for state 2-designs against its t-design counterparts, with $t > 2$[5]. We state the inequality for a state t-design based architecture.

*State t-design Welch bound:*  The choice of the polynomial function for state $t$-designs is presented in Appendix A.4. We state the following inequality for a state in a d-dimensional Hilbert space, $|\psi_i\rangle \in \mathbb{C}^d$ (cf. Appendix B.2 for the derivation):

$$\sum_{1 \leq j,k \leq n} |\langle \psi_j | \psi_k \rangle|^{2t} \geq \frac{n^2}{\binom{d+t-1}{t}} = \frac{n^2}{\frac{(d+t-1)(d+t-2).....d}{t!}} \quad \forall t \in \mathbb{N}. \tag{4}$$

It is clear that for an increasing $t$, the term in the denominator $\frac{(d+t-1)(d+t-2).....d}{t!}$ increases rapidly (as compared to Equation 24 for 2-designs), yielding a vanishing overlap between the state vector summands. This makes the inequality in Equation 4 more stringent for increasing $t$ values.

---

[4]Unitary designs and the difference between unitary and state t-designs are discussed in Appendix A.3.

[5]As mentioned in Appendix A.3, unitary t-designs induce state t-designs when acting on fixed input states. Thus, our data-agnostic approach is also valid when considering state t-designs in the following.

This indicates that higher-order designs ($t > 2$) approximating the Haar measure up to the $t$-th order impose stronger constraints, requiring greater separation between a sequence of state vectors $\{\psi_1, \psi_2, ...., \psi_d\}$ to maintain equality in the bounds. This means that 2-design models have far fewer *degrees of freedom* in building complex quantum circuits. On the contrary, t-designs ($t > 2$) can more efficiently construct higher-order representations capturing more complex interactions between quantum states. Thus, evaluating Welch bounds for 2-designs corroborates its inadequacy for constituting an expressive architecture.

An ansatz forming a 2-design is not expressive as following the Haar measure only up to the second moment constrains how *spread-out* (statistical spread) the unitaries must be, however, does not allow drawing conclusions about how densely it covers $\mathcal{U}(N)$. If one were to define expressivity as the capability of a model to represent all possible set of unitaries, it would be required to go way beyond second moments (for, e.g., Kurtosis, off-diagonal correlations, etc.) to ensure the necessary weights are captured where there is sufficiently dense population. As the moment operator requires tensor products proportional to the moments, this becomes computationally infeasible very quickly. It is not clear how to overcome the curse of dimensionality while proposing a measure that captures expressivity adequately; in fact, most tools from QIT suffer from this very phenomenon, speaking to the difficulty of designing such a metric. Further, there already exists a line of non-unitary VQA research (e.g., Cong et al. (2019); Deshpande et al. (2024)), that needs to be considered as well.

As expressivity is currently the most common metric used to describe the ansatzes, our work employs an alternative approach to analyze the model space, *unrelated to expressivity*, but highly relevant for trainability and training dynamics. Therefore, we focus on local neighborhoods of the model space, in particular, we study how the QNN changes upon perturbation of parameters. This allows analyzing training dynamics and tracking the extent to which the QNN changes during training.

## 4 CHANNEL SENSITIVITY

Comparison between quantum channels is an important topic in Quantum Information Theory. For consistency with respect to, e.g., highly entangling operations, however, operator norms need to be stable under tensor product. Therefore, the diamond norm was proposed in Kitaev (1997) as follows.

**Definition 1** *The **diamond norm** of a superoperator $T$ is defined as (Harrow & Low, 2009; Kitaev et al., 2002).*

$$\|T\|_\diamond = \sup_d \|T \otimes id_d\|_\infty = \sup_d \sup_{X \neq 0} \frac{\|(T \otimes id_d)X\|_1}{\|X\|_1} \tag{5}$$

Here, $\otimes$ denotes the tensor product and $id_d$ the identity channel on $d$ dimensions, with $d \leq 2^n$, and $n$ as the number of qubits. The $\|X\|_1$ in this context is the *Trace norm* or *Schatten 1-norm*, which is defined as $\|X\|_1 = Tr\sqrt{X^\dagger X}$. The diamond norm is defined for any superoperator $T$, wherein, superoperators are defined as the set of linear maps acting on a vector space of linear operators. For a valid quantum channel $\mathcal{E}_1$ (a completely positive and trace preserving (CPTP) matrix), the diamond norm is strictly upper bounded by $1$ and lower bounded by $0$. When using the diamond norm to measure the distance between two CPTP quantum channels $\mathcal{E}_1$ and $\mathcal{E}_2$, the channels are subtracted. The resulting matrix is not necessarily CPTP, resulting in a revised upper bound of 2.

Operationally, the diamond norm measures the maximum distinguishability between the output states of the two maps under any input state, i.e., for any input state we will be *less or equally able to distinguish* the output. A value of $0$, means that they are indistinguishable, whereas a value of 2 means they are perfectly distinguishable. Computing the diamond norm is non-trivial, however, can be reduced to a semi-definite program (Watrous, 2009; 2013).

We apply the diamond norm to evaluate the distinguishability of a parametrized unitary $U(\boldsymbol{\vartheta})$, representing a quantum channel, from the unitary $U(\boldsymbol{\vartheta} + \boldsymbol{\delta})$, where $\boldsymbol{\delta}$ is a perturbation of $\boldsymbol{\vartheta}$, as follows.

**Definition 2** *We define the **channel sensitivity** as the distinguishability of the quantum channel from itself upon small perturbation of $\boldsymbol{\vartheta}$ by $\boldsymbol{\delta}$.*

Table 1: Employed ansatzes

| COMPONENT | CONFIGURATIONS |
|---|---|
| Parameterized | $[R_X, R_Y, R_Z, R_X R_Y, R_Y R_Z, R_X R_Z, R_X R_Y R_Z]$ |
| Entangling | [CNOT, CZ, CNOT CZ] |

$$cs_U(\boldsymbol{\vartheta}, \boldsymbol{\delta}) = \|U(\boldsymbol{\vartheta}) - U(\boldsymbol{\vartheta} + \boldsymbol{\delta})\|_\diamond \tag{6}$$

We can provide an upper bound on the channel sensitivity through Taylor expansion, which is shown in Equation 7. Therefore, we establish a direct dependence of the distinguishability of the channels to the sum of changes applied[6]. The bound holds if the Hermitian generators of the trainable gates are unitary as well, which is relevant for HEAs, as the trainable gates are exponentiations of the $X$, $Y$, and $Z$ Pauli gates. The precision of the bound depends on the magnitude of $\boldsymbol{\delta}$, however, considering that the models are trained iteratively in small steps, we expect the bound to hold.

$$\|U(\boldsymbol{\vartheta}) - U(\boldsymbol{\vartheta} + \boldsymbol{\delta})\|_\diamond = \|U(\boldsymbol{\vartheta}) - (U(\boldsymbol{\vartheta}) + \sum_{j=1}^{dim(\boldsymbol{\delta})} \delta_j \frac{\partial U(\boldsymbol{\vartheta})}{\partial \vartheta_j} + O(\boldsymbol{\delta}^2))\|_\diamond$$
$$\approx \| - \sum_{j=1}^{dim(\boldsymbol{\delta})} \delta_j \frac{\partial U(\boldsymbol{\vartheta})}{\partial \vartheta_j}\|_\diamond \le \frac{\sum_{j=1}^{dim(\boldsymbol{\delta})} |\delta_j|}{2} \tag{7}$$

Unless the feature encoding is trained as well, the upper bound on distinguishability cannot be improved by including data. Intuitively, due to the calculation of the diamond being based on the maximum distinguishability from any input state, it includes any possible unitary data encoding as well. Mathematically, this property follows from the unitary invariance of the Schatten 1-norm.

Our bound shows that (assuming small parameters updates), there is a direct relationship between the number of parameters and the distinguishability of the operation. In particular, for small models, which are used extensively at the moment, it can be expected that subsequent unitaries during training are hardly distinguishable, hindering effective training. This is, to the best of our knowledge, the first result attempting to establish a connection between ansatz and trainability issues. To strengthen our results, we want to, in the following, support our bounds through numerical experiments.

## 5 EXPERIMENTAL SETUP

**Ansatzes**  We consider layered architectures in our experiments, each consisting of trainable *parameterized* and fixed *entangling* components. For the first, we use rotations around the x- ($R_X$), y- ($R_Y$), and z-axis ($R_Z$) of the Bloch sphere, and the controlled-NOT (CNOT) and controlled-Z (CZ) gates for the latter[7], constituting a widely-used gate set for HEAs. For a list of parameterized and entangling components, we refer to Table 1, and we run our experiments using all combinations.

We do not permute the operators, due to considerations on the expressivity per qubit. That is, if all rotations are applied, all permutations of $R_X(\alpha)R_Y(\beta)R_Z(\gamma)$, with $\alpha, \beta, \gamma \in [0, 2\pi]$ span the $SU(2)$, the set of unitary operations on one qubit. Therefore, permuting the operations does not affect expressivity per layer. Similarly, two rotation operations per layer explore a subspace of the $SU(2)$ owing to the Lie-algebraic closure relations, e.g., $R_X(\alpha)R_Y(\beta)$ spans the upper half of the Bloch sphere, as does $R_Y(\alpha)R_X(\beta)$. Any permutation allows the same operations per qubit, which is why we deem all permutations as equal.

**Perturbation**  For the experiments, we uniformly sample parameters in range $[0, 2\pi]$ and choose a random 95% of them to perturb their values by $\pm t\%$ (in our case $[0.1, 0.5, 1]$). These values were in

---

[6]For the precise mathematical derivation, we refer to Appendix C.
[7]We refer to Appendix A.1 for an overview of the operations.

no way chosen randomly, rather, by training QNNs and collecting summary statistics on parameter updates, we found that these are the ranges where updates are performed. Then, we evaluate the channel sensitivity with the two obtained unitaries. We take 100 samples per parameter involved in the circuit and consider the distribution of the channel sensitivity we obtain for further analysis.

**Training** Moreover, we want to compare the channel sensitivity for random perturbations and training runs. Therefore, we train binary classification models on the two largest classes of the wine and breast cancer datasets and use PCA to reduce the features to $2^n$, with $n$ as the number of qubits. Then, the features are normalized and encoded into the amplitudes of the input state. The measurement is taken in the z-basis of the first qubit, and we use the mean squared error as the loss.

To account for favorable parameter initializations, we train each architecture 50 times. We use 150 iterations, and monitor convergence and parameter changes. We use the Adam optimizer (Kingma & Ba, 2015) from Pennylane with default parameters (step size: 0.01, beta 1: 0.9, beta 2: 0.99, epsilon: 1e-08). Further, we calculate the channel sensitivity in every iteration, to get an operationally meaningful analysis of how much the model changes. This allows evaluating whether the model changes more when updated based on the loss function, rather than by just randomly permuting parameters. We display aggregated results using confidence intervals with a 95% confidence level.

**Qubits** Due to the computational complexity involved in solving the diamond norm, we run our experiments from one to four qubits and one to five layers. Unfortunately, due to matrix dimensions, it is challenging to scale the diamond norm any further. Nonetheless, the results are relevant, considering that they (1) provide a first step at analyzing small scale architectures, and (2) have direct implications for training dynamics that can be expected in the NISQ era.

Today's QNNs use shallow circuits with a depth in $O(\log n)$ and a local cost function, as they can be shown to be trainable (Cerezo et al., 2021b). Even if favorable loss landscapes were found for deeper circuits, adding layers results in noisier loss estimates, prohibiting effective learning. Moreover, scaling in number of qubits is limited due to quantum hardware limitations, or, when the models are simulated classically, to the curse of dimensionality. Therefore, many recent works proposing QML to solve a particular problem use few qubits, e.g., Blance & Spannowsky (2021), with two qubits, or Yano et al. (2020) with three and four qubits.

Further, our main goal is to take initial steps in quantifying and characterizing QNN model behavior. As has been mentioned before, due to the power of classical ML, we are unlikely to find an advantage in QML in the near term (Schuld & Killoran, 2022). Therefore, it would be important to focus on more fundamental questions, such as "how can we exploit quantum mechanics for ML purposes?", to explore the potential of quantum computing for data processing. In that sense, understanding how quantum channels change upon parameter updates is a foundational topic in ansatz design.

**Framework** Our experiments are implemented using the Python programming language. We use Pennylane (Bergholm et al., 2022) for obtaining the unitary transformations and QuTIP (Johansson et al., 2012; 2013) for calculating the diamond norm. Unfortunately, a diamond norm computation between two operators $\|A - B\|_\diamond$ with a finitely-large overlap $A^\dagger B \approx \mathbb{I}$, yields numerical instabilities with the QuTIP package. To account for these numerical issues, the operator can be multiplied with a global phase, which fixes a gauge (cf. *Gauge-Fixing*, in Appendix F).

## 6 EXPERIMENTAL RESULTS

We run extensive numerical experiments to validate our bounds. First, we randomly perturb parameters, then we compare them to the channel sensitivity we observe during actual training runs. Initially, we verify the assumptions about the magnitude of parameter changes. In training runs, the mean parameter change per update in the first 10 iterations is maximum $1.4\%$, with a mean and standard deviation of $0.8\% \pm 0.4\%$, but it quickly decreases. We visualize this in Figure 1a, with the iterations on the x-axis, and the mean change in parameters on the y-axis. Architectures with the same number of qubits are aggregated and show very narrow confidence intervals. Further, it is also visible that changes in multi-qubit systems tend to be even smaller (less than two orders of magnitude). Moreover, we observe that almost all parameters are updated in every iteration, which is shown in Figure 1b, where the y-axis shows the percentage of parameters changed.

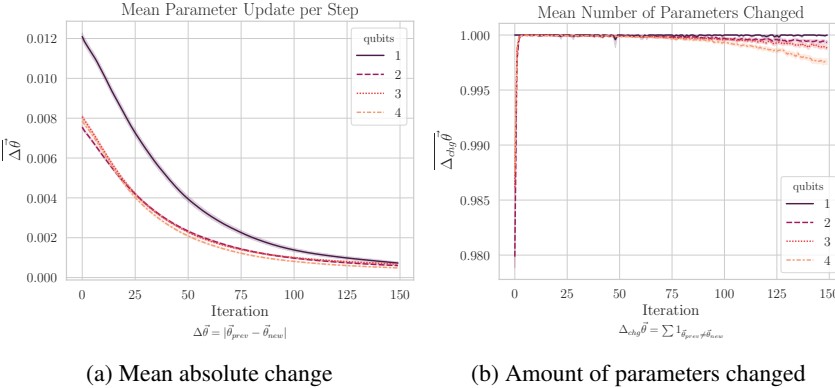

(a) Mean absolute change  (b) Amount of parameters changed

Figure 1: Training parameter updates

## 6.1 RANDOM PERTURBATION

Based on these observations, we run the random perturbation experiments by changing 95% of parameters by 1%, 0.5%, and 0.1%. We visualize the results in Figure 2, showing the depth of the circuit on the x-axis and the obtained channel sensitivity on the y-axis. Per our bounds, the channel sensitivity strongly depends on the number of parameters in the circuit, hence, we visualize the results based on the number of parameters per layer.

The experiments show the same patterns as the bounds predict. In particular, fixing the qubits and parameters per layer while increasing the depth of the circuit results in a bigger channel sensitivity. The same can be observed when fixing the number of layers and increasing the qubits. We observe that increasing channel sensitivity is more prevalent for larger systems than for deeper circuits, e.g, architectures with 2 qubits, 5 layers and 3 parameters have a smaller channel sensitivity than architectures with 3 qubits, 5 layers and 2 parameters, despite having the same number of parameters.

The behavior is expected for ansatzes covering large parts of the search space, as the Hilbert space of a larger system is significantly bigger, scaling as $2^n$. This allows more independent search directions, therefore, potentially more changes in channel upon update. Deeper circuits can explore larger parts of the Hilbert space too, however, their expressivity stagnates when achieving overparameterization (Larocca et al., 2023). In this regime, more parameters will not result in more independent search directions, hence, the difference in channel sensitivity is expected to be smaller.

Further, it can be seen that there is a substantial degree of variability in the channel sensitivity. In the majority of experiments, the channels are hardly distinguishable, however, many outliers can be observed. This property could be particularly relevant for warm-starting (Puig et al., 2024), i.e., starting training from regions with high channel sensitivity might lead to smoother optimization.

## 6.2 TRAINING UPDATES

Additionally, we want to consider how different the quantum channels are when updated based on loss and optimizer. In particular, it could be possible that changing the parameters during training, changes the underlying channel a lot more than perturbing in random directions. Upon analysis of the results, we can confirm the validity of the Taylor expansion even in early stages of iteration, i.e., in the $45,500$ models we trained, the bound holds for every single parameter update taken.

We plot the channel sensitivity in Figure 3 for different layers, qubits and parameters per layer. We can observe that the confidence intervals are very narrow, i.e., even though we collect data from 50 different runs for every model, the observable channel sensitivity hardly varies. Further, the channel sensitivity is substantially smaller than the bound, speaking to the severity of the issue. The discrepancy grows in particular with the number of qubits, which is visualized in Figure 4 in Appendix D. While we cannot observe the magnitude of channel sensitivity that the bound predicts, the pattern of scaling in the number of parameters can be observed. This is in particular visible in Figure 3 for early training stages with larger parameter updates, however, it vanishes in later

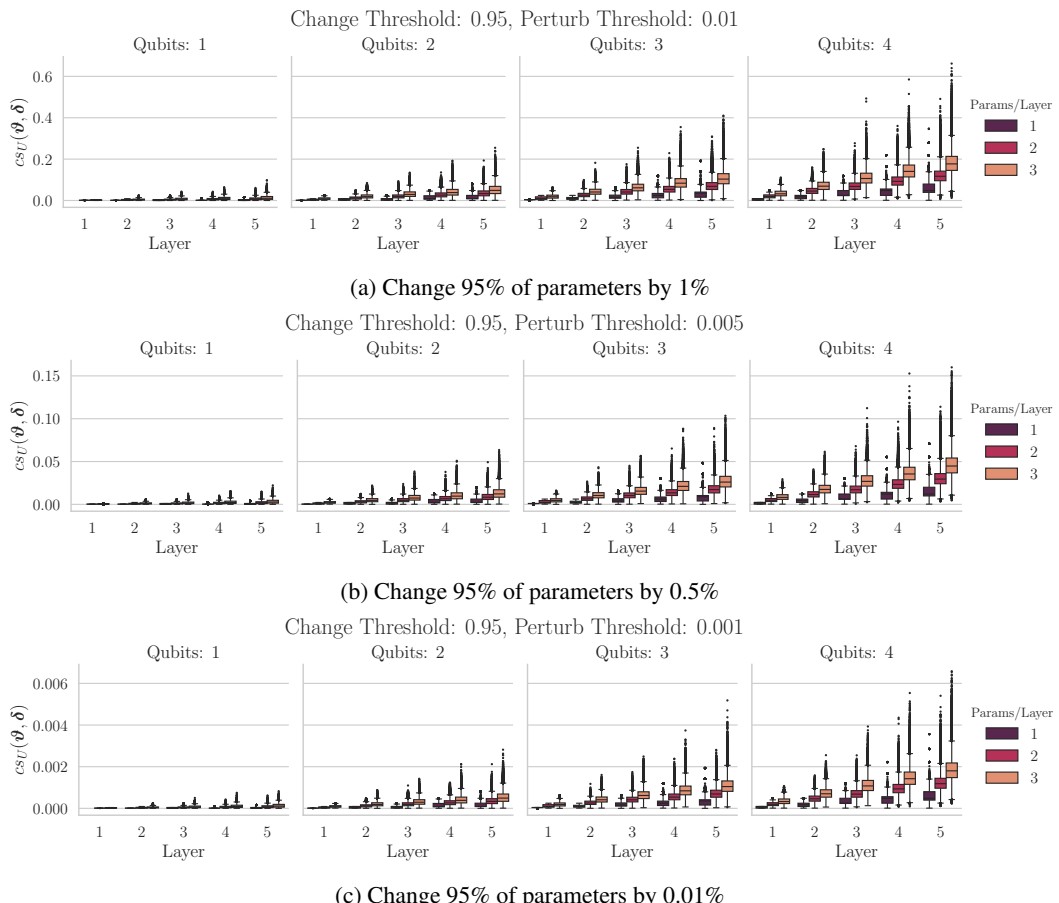

Figure 2: Channel sensitivity for random perturbations

stages, when the updates are very small. Further, to showcase robustness with respect to the feature encoding (and different parameter updates that may result thereof), we run our experiments with angle encoding as well and verify that the magnitude of channel sensitivity does not change. We refer to Figure 5 in Appendix E for more information.

## 7  DISCUSSION

Our bound establishes a direct dependence of the distinguishability of the channels during QNN training to the number of parameters. That is, the *maximum distinguishability of two output states* of the quantum channels, scales with (1) the magnitude of changes and (2) the number of parameters. As iterative updates with small learning rates are applied, which is independent of the number of parameters in the circuit, the dependence is largely dominated by the number of parameters.

This could significantly contribute to the trainability issues of today's QNNs. In particular, for reasonably sized models, the channels may be distinguishable in early training stages, but hardly distinguishable when fine-tuning the ansatz later. Comparison of our bound to the channel sensitivity obtained in training runs reveals that during training, the channels are even less distinguishable. While the channel sensitivity for random permutation is still considerably lower than our bounds would predict, a lot more variation can be observed.

This confirms our initial motivation of studying the model space independently of the loss and data, as these are global properties of ansatzes. In particular, we want to draw attention to the remarkable similarities to the BP phenomenon. Adapting a series of composition of maps viewpoint (Larocca et al., 2023), our work argues that the unitaries in the unitary space of the considered architectures

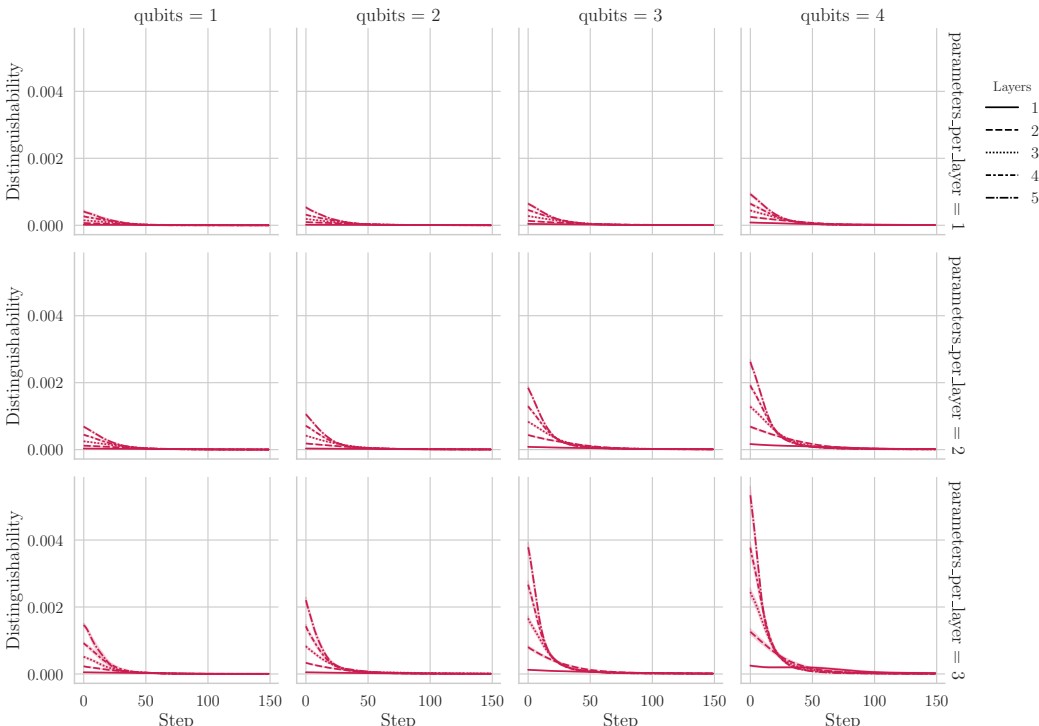

Figure 3: Channel sensitivity during training runs

do not change significantly upon parameter update in the local neighborhood of the parameter space. Thus, from small channel distinguishability, it follows that the quantum states in the Hilbert space will not differ significantly, as will the expectation values (which is a consequence of the diamond norm formulation), which are directly related to the loss. Roughly speaking, this behavior is mirrored in the BP phenomenon. While we do not argue for strict equivalence, the similarities hint towards two perspectives on similar trainability issues. Therefore, this phenomenon might not only be tied to the exponential Hilbert space, but may be an artifact of the architectures themselves.

Therefore, it seems that the architectures significantly contribute to the untrainable models we observe today, independently of data and loss. Our findings have implications to start a more thorough investigation of the model space of the ansatzes that are currently used, and, ultimately, stress the need for good initialization and warm-starting. It is, however, non-obvious how to find such starting points without trial-and-error.

## 8  CONCLUSION

Altogether, our work provides a first step at exploring the model space of QNNs. In particular, we provide a picture of the local neighborhood of the model space of an ansatz, which suggest that the channels, at least for small scale circuits, do not differ significantly upon parameter update. Our results for random perturbations reveal that there is a substantial degree of variability of the channel sensitivity depending on the neighborhood, hence, stress the need to study the model space more thoroughly. This could reveal valuable information for parameter initialization or warm-starting.

Considering that QNNs were designed for NISQ technology, shallow, small-scale circuits due to hardware limitations, the community will be limited to such small instances in the near-term. Therefore, our findings have direct implications on the meaningfulness of iteratively optimizing such circuits. While it is true that the models can be scaled to work for larger circuits, we have overwhelming evidence through works on BPs, that this is inherently difficult. Together with results on classical simulability and dequantizability in the absence of BPs, this work extends the evidence that we need a paradigm shift in Variational Quantum Computing or even QML altogether.

## ACKNOWLEDGMENTS

The authors would like to thank Adrián Pérez-Salinas for feedback on earlier versions of this work. This work has been partially funded through the Themis project (Trustworthy and Sustainable Code Offloading), Austrian Science Fund (FWF): Grant-DOI 10.55776/PAT1668223, the Standalone Project Transprecise Edge Computing (Triton), Austrian Science Fund (FWF): P 36870-N, and by the Flagship Project HPQC (High Performance Integrated Quantum Computing) # 897481 Austrian Research Promotion Agency (FFG).

## REPRODUCIBILITY STATEMENT

The code and results for the experiments are publicly available in a GitHub repository[8].

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

## A ADDITIONAL BACKGROUND

### A.1 QUANTUM COMPUTING

Quantum Computing (QC) works with quantum bits (qubits). While classical bits take either the value 0 or 1, qubits are in a so-called *superposition* of the two states, meaning they are in both states at once, which is shown mathematically in Equation 8, where $\alpha$ and $\beta$ are called *probability amplitudes*. We can do calculations with the superposed qubit, however, once we read out results, we have to take a *measurement*, where the qubit will collapse into state $|0\rangle$ with probability $|\alpha|^2$, and into state $|1\rangle$ with probability $|\beta|^2$. Due to this inherent non-determinism in QC, executions are always done multiple times.

$$|\psi\rangle = \alpha |0\rangle + \beta |1\rangle \quad \text{s.t. } |\alpha|^2 + |\beta|^2 = 1 \quad \alpha, \beta \in \mathbb{C} \tag{8}$$

A qubit is characterized by four numbers (the real and imaginary parts of $\alpha$ and $\beta$), which, when translated into polar coordinates, gives a convenient geometrical representation on the so-called *Bloch sphere*. This plays a significant part in today's QNNs, as the trainable gates are rotations of the individual qubits around the x-, y- and z-axis of this sphere.

Multiple qubits form a *quantum register*, and its associated quantum state represents $2^n$ states at once, where $n$ is the number of qubits, which can make QC very powerful. A quantum state can be

*entangled*, meaning that individual qubits can be correlated, such that an action on one qubit affects the ones it is entangled with as well.

Quantum states are manipulated through *unitary transformations* or *quantum gates* ($U^\dagger U = \mathbb{I}$, where $U^\dagger$ is the adjoint of $U$). Unitary transformations are linear and preserve vector lengths, i.e., applying a unitary to a quantum state ensures that the squared magnitudes of the probability amplitudes still sum up to one. Every unitary is generated by a *Hermitian generator* ($H = H^\dagger$), i.e., $U = e^{-iH}$. We describe the action on an initial quantum state as a so-called *unitary evolution*, which may involve one or more unitary operations (unitary matrices form a group, hence, are closed under multiplication). We also refer to this as a *quantum channel* in the following. While the term quantum channel encompasses unitary evolutions, it is a broader term, meaning it can also describe more general dynamics of quantum systems, such as decoherence or noise. It is defined as a linear map that is completely positive and trace preserving (Watrous, 2018).

Among the most fundamental operations in QC, are the so-called Pauli matrices, which are Hermitian matrices shown in Equation 9. $\sigma_x$ applies a NOT (bit-flip) operation ($|0\rangle \rightarrow |1\rangle$, $|1\rangle \rightarrow |0\rangle$), $\sigma_y$ changes the phase and bit ($|0\rangle \rightarrow i|1\rangle$, $|1\rangle \rightarrow -i|0\rangle$), and $\sigma_z$ applies a phase-flip ($|0\rangle \rightarrow |0\rangle$, $|1\rangle \rightarrow -1|1\rangle$).

$$\sigma_x = \begin{bmatrix} 0 & 1 \\ 1 & 0 \end{bmatrix} \quad \sigma_y = \begin{bmatrix} 0 & -i \\ i & 0 \end{bmatrix} \quad \sigma_z = \begin{bmatrix} 1 & 0 \\ 0 & -1 \end{bmatrix} \tag{9}$$

Taking the Pauli matrices as generators, we can rotate a qubit around the x-, y- and z-axis around the Bloch sphere, arriving at the definition of the $R_X$, $R_Y$ and $R_Z$ rotation in Equation 10.

$$\mathrm{R_X}(\theta) = e^{-i\frac{\theta}{2}\sigma_x} \quad R_Y(\theta) = e^{-i\frac{\theta}{2}\sigma_y} \quad R_Z(\theta) = e^{-i\frac{\theta}{2}\sigma_z} \tag{10}$$

Further, we define the two entangling operations we use, CNOT (or CX) and CZ, in Equation 11. The operators entangle two qubits $i$ and $j$, where $i$ is referred to as the *control* qubit and $j$ is the *target* qubit. In general terms, the entangling gates apply $\sigma_x$ or $\sigma_z$ to the target qubit if the control is in state $|1\rangle$.

$$\mathrm{CNOT} = \begin{bmatrix} 1 & 0 & 0 & 0 \\ 0 & 1 & 0 & 0 \\ 0 & 0 & 0 & 1 \\ 0 & 0 & 1 & 0 \end{bmatrix} \quad \mathrm{CZ} = \begin{bmatrix} 1 & 0 & 0 & 0 \\ 0 & 1 & 0 & 0 \\ 0 & 0 & 1 & 0 \\ 0 & 0 & 0 & -1 \end{bmatrix} \tag{11}$$

## A.2 Inner-product & overlap between vectors

An inner-product space is defined as a vector space $V$ over a field $\mathbb{F}$ (typically $\mathbb{R}$ or $\mathbb{C}$) endowed with an inner-product map $\langle .|. \rangle$:

$$\langle .|. \rangle : V \times V \rightarrow \mathbb{F} \tag{12}$$

The inner-product operation is a generalization of the dot-product between vectors and measures the **overlap** or **coherence** between vectors sampled from a vector space.

## A.3 Unitary groups and the Haar Measure

We will provide key information on unitary groups and the Haar measure in this section. For an excellent tutorial on the topic, we refer to Mele (2024), which we also base this section on. We begin by defining the unitary group $\mathcal{U}(d)$ and the special unitary group $S\mathcal{U}(d)$, which are key concepts when studying quantum computing, as follows.

**Definition 3** *For $d \in \mathbb{N}$, the **unitary group** $\mathcal{U}(d)$ is the group of isometries of $d$-dimensional complex Hilbert space $\mathbb{C}^d$. These are canonically identified with the $d \times d$ unitary matrices ($\mathbb{C}^{d \times d}$):*

$$\mathcal{U}(d) = \left\{ \mathcal{V} \in \mathbb{C}^{d \times d} | \mathcal{V} \cdot \mathcal{V}^\dagger = \mathcal{V}^\dagger \cdot \mathcal{V} = \mathbb{I}_d \right\} \tag{13}$$

The unitary group can be decomposed as $\mathcal{U}(d) = S\mathcal{U}(d) \times U(1)$. Here, the subgroup $S\mathcal{U}(d)$ is called the *special unitary* group, while $z = e^{i\theta} \; \forall z \in U(1)$ is a phase restricted on a unit-circle.

**Definition 4** *For $d \in \mathbb{N}$, the **special unitary group** $S\mathcal{U}(d) \subset \mathcal{U}(d)$ are a group of $d \times d$ unitary matrices ($\mathbb{C}^{d \times d}$) that have a unit determinant:*

$$S\mathcal{U}(d) = \left\{ \mathcal{G} \in \mathcal{U}(d) \mid det(G) = 1 \right\} \tag{14}$$

The Haar measure forms a uniform probability distribution over sets of unitary matrices, in fact, it is unique for compact groups, such as the $\mathcal{U}(d)$.

**Definition 5** *(Mele, 2024) We define the **Haar Measure** on the $\mathcal{U}(d)$, as the left and right invariant probability measure $\mu_H$ over the group. That is, for all integrable functions $f$ and $\forall V \in \mathcal{U}(d)$:*

$$\int_{\mathcal{U}(d)} f(U) d\mu_H(U) = \int_{\mathcal{U}(d)} f(UV) d\mu_H(U) = \int_{\mathcal{U}(d)} f(VU) d\mu_H(U) \tag{15}$$

Thus, the Haar measure assigns an invariant volume measure to subsets of locally compact topological groups. Due to properties of a probability measure, it holds that $\int_S d\mu_H(U) \geq 0, \forall S \subseteq \mathcal{U}(d)$ and $\int_{\mathcal{U}(d)} 1 d\mu_H(U) = 1$. It is therefore possible to set the expectation value with respect to the probability measure equal to the integral over the Haar measure as $\mathbb{E}_{U \sim \mu_H}[f(U)] = \int_{\mathcal{U}(d)} f(U) d\mu_H(U)$. This leads to the definition of the moment operator of the Haar measure.

**Definition 6** *(Mele, 2024) For all operators $O$, the **t-th moment operator** based on probability measure $\mu_H$ is defined as follows.*

$$M_{\mu_H}^{(t)}(O) = \mathbb{E}_{U \sim \mu_H}[U^{\otimes t} O U^{\dagger \otimes t}] \tag{16}$$

**Definition 7** *(Mele, 2024) A distribution $v$ over a set of unitaries $S \subseteq \mathcal{U}(d)$ is a **unitary t-design** if and only if the following holds for all operators $O$.*

$$\mathbb{E}_{V \sim v}[V^{\otimes t} O V^{\dagger \otimes t}] = \mathbb{E}_{U \sim \mu_H}[U^{\otimes t} O U^{\dagger \otimes t}] \tag{17}$$

State t-designs are weaker versions of unitary t-designs in the sense that unitary t-designs induce state t-designs when they act on a fixed input state. These are defined as follows.

**Definition 8** *(Mele, 2024) A **state t-design** is a probability distribution $\eta$ over a set of states $S$ if and only if the following holds.*

$$\mathbb{E}_{|\psi\rangle \sim \eta}[|\psi\rangle \langle \psi|^{\otimes k}] = \mathbb{E}_{|\psi\rangle \sim \mu_H}[|\psi\rangle \langle \psi|^{\otimes k}] \tag{18}$$

Further, we define an equivalent definition of state t-designs as follows, which will be used as well.

**Definition 9** *(Hoggar, 1982) Let $\mathbb{C}^d$ denote a d-dimensional Hilbert space with orthonormal basis set $\{|n_k\rangle\}_{k=1}^d$. Owing to their redundancy under global phase transformations and normalization, the quantum states in this space correspond to points in the complex-projective space $\mathbb{CP}^{d-1}$ (Bengtsson & Życzkowski, 2006). Thus, we define the nontrivial complex-projective t-design as a set of states $S \subsetneq \mathbb{CP}^{d-1}$, sampled according to some probability measure $\mu$, satisfying,*

$$\mathbb{E}_{|\psi\rangle \in S} f(|\psi\rangle) = \int_{\mathbb{CP}^{d-1}} f(|\psi\rangle) d\psi. \tag{19}$$

*Where, $f(|\psi\rangle)$ is a polynomial of at most degree $t$ in its amplitudes and complex-amplitudes of $|\psi\rangle$ respectively. The canonical measure $d\psi$ defined on the set of such quantum states, is the unique unit-normalized volume measure invariant under unitary group action $\mathcal{U}(d)$.*

Often it is not strictly required to have an exact t-design, but rather a distribution close to a t-design, which is termed $\epsilon$-approximate t-design.

### A.4 Choice of polynomial function for state t-designs:

The function $f(|\psi\rangle)$ is a polynomial of at most degree $t$. For example, 2-designs, correspond to the average value of a quadratic (second-degree polynomial) function. Hence, a common choice of state 2-design polynomial functions is the overlap (cf. Appendix A.2 for definition) between a collection of states $\{\psi_i\}_{1 \leq i \leq d}$ in the Hilbert space $|\langle\psi_j|\psi_k\rangle|^2$. Similarly, for an even $t > 2$, a canonical choice of the polynomial function $f$ is $|\langle\psi_j|\psi_k\rangle|^{2t}$.

## B Welch bounds and t-designs

**Theorem 1:** **(Welch bounds)** Let $n \geq d$. If $\{v_i\}_{1 \leq i \leq n}$ is a sequence of unit vectors in $\mathbb{C}^d$, then,

$$\max_{1 \leq j,k \leq n, j \neq k} |\langle v_j|v_k\rangle|^{2t} \geq \frac{1}{n-1}\left[\frac{n}{\binom{d+t-1}{t}} - 1\right], \ \forall t \in \mathbb{N}, \tag{20a}$$

$$\text{implies,}$$

$$\sum_{1 \leq j,k \leq n} |\langle v_j|v_k\rangle|^{2t} \geq \frac{n^2}{\binom{d+t-1}{t}} \forall t \in \mathbb{N}. \tag{20b}$$

The combinatorial (binomial) factor featuring in the denominator of Equation 20a and Equation 20b can be expanded in terms of the factorials:

$$\binom{d+t-1}{t} = \frac{(d+t-1)!}{(d-1)! \ t!} \tag{21}$$

Here ! corresponds to the factorial symbol, i.e. $n! = n(n-1)(n-2)...1$.

### B.1 Simplification of Equation 20a:

Substituting Equation 21 into Equation 20a yields the following.

$$\max_{1 \leq j,k \leq n, j \neq k} |\langle v_j|v_k\rangle|^{2t} \geq \frac{1}{n-1}\Big[\frac{n}{\frac{(d+t-1)!}{(d-1)! \ t!}} - 1\Big] = \frac{1}{n-1}\Big[\frac{n}{\frac{(d+t-1)(d+t-2)....d}{t!}} - 1\Big] \tag{22}$$

Here in the denominator, we have used the factorial property, $(d+t-1)!/(d-1)! = (d+t-1)(d+t-2)...d(d-1)!/(d-1)! = (d+t-1)(d+t-2)...d$.

### B.2 Simplification of Equation 20b:

Using Equation 21 it follows that:

$$\sum_{1 \leq j,k \leq n} |\langle v_j|v_k\rangle|^{2t} \geq \frac{n^2}{\frac{(d+t-1)(d+t-2)...d}{t!}}. \tag{23}$$

For the special case, $t = 2$, we get the inequality for **2-designs**,

$$\sum_{1 \leq j,k \leq n} |\langle v_j|v_k\rangle|^4 \geq \frac{n^2}{\binom{d+1}{2}} = \frac{n^2}{(d+1)d/2}. \tag{24}$$

## C Channel Sensitivity Bound

### C.1 Preliminaries

Before deriving the bound, we want to state some preliminaries for partial derivatives of parameterized unitaries. Equation 25 shows the partial derivative of a circuit (Holmes et al., 2022) which utilizes the convenient representation of a unitary in terms of its generator. The partial derivative essentially splits the original unitary in two parts, and we define the fractions in Equation 26.

$$\partial_j U(\boldsymbol{\vartheta}) = -\frac{i}{2} U_{j+1 \to dim(\boldsymbol{\vartheta})} H_j U_{1 \to j} \tag{25}$$

$$
\begin{aligned}
U_{l \to m}(\boldsymbol{\vartheta}) &= \prod_{j=l}^{m} V_j(\vartheta_j) W_j \\
&= \prod_{j=l}^{m} e^{-i\frac{\vartheta_j}{2} H_j} W_j
\end{aligned}
\tag{26}
$$

## C.2 DERIVATION OF THE BOUND

We can now analyze the difference in diamond norm upon slight perturbation of parameters in Equation 27. We use a Taylor expansion and truncate after the first order. We arrive at Equation 28 after applying the partial derivative from Equation 25. We use the triangle inequality and homogeneity, which are both necessary conditions for matrix norms (Horn & Johnson, 1985), to arrive at Equation 29 and Equation 30 respectively. Arriving at Equation 31 is non-trivial, as, in general, it is not guaranteed that multiplying hermitian and unitary matrices results in a unitary. However, HEAs use $R_X$, $R_Y$ and $R_Z$ rotations with generators $X$, $Y$ and $Z$ respectively, which are known to be hermitian and unitary. Since unitaries form a group that is closed under multiplication, it forms a valid quantum channel, which evaluates to 1(Watrous, 2018, Proposition 3.44).

$$\|U(\boldsymbol{\vartheta}) - U(\boldsymbol{\vartheta} + \boldsymbol{\delta})\|_\diamond = \|U(\boldsymbol{\vartheta}) - \left(U(\boldsymbol{\vartheta}) + \sum_{j=1}^{dim(\boldsymbol{\delta})} \delta_j \frac{\partial U(\boldsymbol{\vartheta})}{\partial \vartheta_j} + O(\boldsymbol{\delta}^2)\right)\|_\diamond \tag{27}$$

$$\approx \|U(\boldsymbol{\vartheta}) - (U(\boldsymbol{\vartheta}) + \sum_{j=1}^{dim(\boldsymbol{\delta})} \frac{\partial U(\boldsymbol{\vartheta})}{\partial \theta_j} \delta_j)\|_\diamond \tag{28}$$

$$= \| - \sum_{j=1}^{dim(\boldsymbol{\delta})} \frac{\partial U(\boldsymbol{\vartheta})}{\partial \theta_j} \delta_j\|_\diamond \tag{29}$$

$$= \| - \sum_{j=1}^{dim(\boldsymbol{\delta})} -\frac{i}{2} U_{j+1 \to dim(\boldsymbol{\delta})} H_j U_{1 \to j} \delta_j\|_\diamond \tag{30}$$

$$\leq \sum_{j=1}^{dim(\boldsymbol{\delta})} \|\frac{i}{2} U_{j+1 \to dim(\boldsymbol{\delta})} H_j U_{1 \to j} \delta_j\|_\diamond \tag{31}$$

$$= \sum_{j=1}^{dim(\boldsymbol{\delta})} |\frac{i}{2} \delta_j| \|U_{j+1 \to dim(\boldsymbol{\delta})} H_j U_{1 \to j}\|_\diamond \tag{32}$$

$$= \frac{\sum_{j=1}^{dim(\boldsymbol{\delta})} |\delta_j|}{2} \tag{33}$$

## D COMPARISON OF EMPIRICAL CHANNEL SENSITIVITY AND BOUND

We compare the empirical channel sensitivity with the predicted bound in Figure 4. We add this figure to visualize that experimentally we can observe a large discrepancy, although the magnitude of this discrepancy makes it hard to read much more from the plot. We also would like to remark that one should be careful in drawing the conclusion that the bound is thus loose, as there might still be a point in the model space where taking such a step could achieve the bound. Our extensive experiments, however, reveal that this will likely not be the case in large areas of the model space.

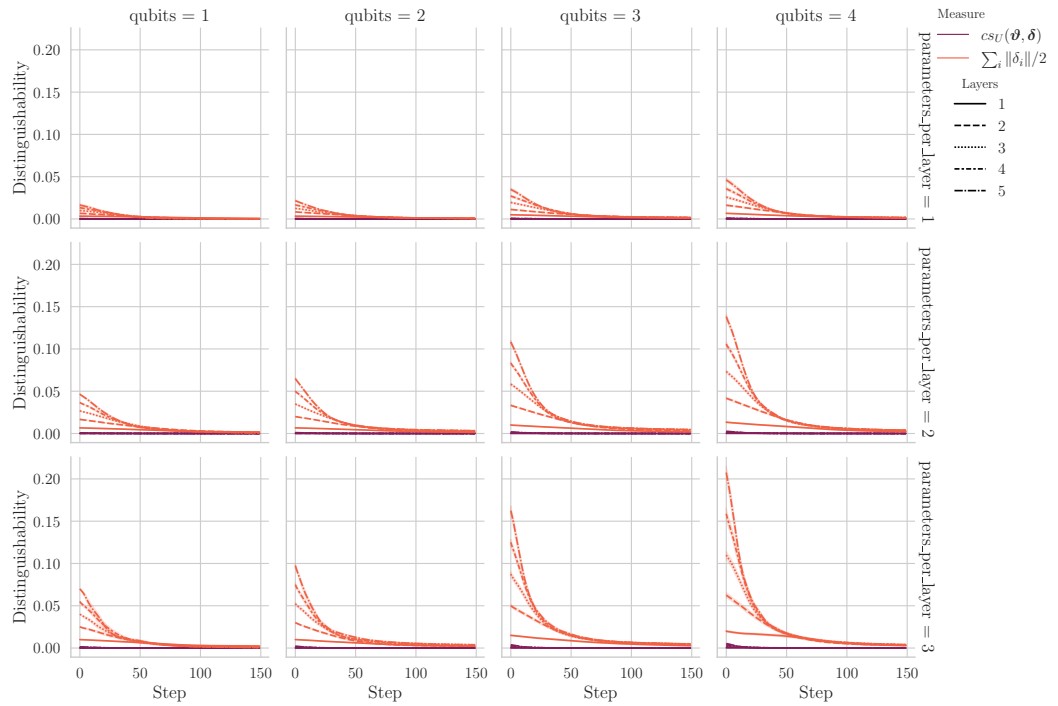

Figure 4: Comparison $\frac{\sum_j |\delta_j|}{2}$ and channel sensitivity

## E  ANGLE ENCODING

We visualize the channel sensitivity when encoding the data with angle embedding. We run these experiments to demonstrate robustness of the channel sensitivity with respect to the changed parameter updates that may occur due the different input states. While the channel sensitivity slightly varies, we can observe that its magnitude does not differ when comparing it to the results using amplitude embedding (we observe a maximum distinguishability of approximately $0.5\%$ in both cases).

## F  GAUGE FIXING

Considering the numerical instabilities whenever $A^\dagger B \approx \mathbb{I}$, we apply a Gauge-Fixing algorithm. In particular, we adjust the global phases of the two unitaries, which ensures numerical stability without affecting the diamond norm.

---

**Algorithm 1** Gauge-fixed diamond norm between difference of quantum channels

---

1: **function** DIAMOND NORM($\|\boldsymbol{A} - \boldsymbol{B}\|_\diamond$)
2:     **if** $\boldsymbol{A}^\dagger \boldsymbol{B} \approx \mathbb{I}$ **then**
3:         $\boldsymbol{z} \in \mathbb{C} \leftarrow \text{Tr}(\boldsymbol{A}^\dagger \boldsymbol{B})$
4:         $\vartheta^* \in \mathbb{R} \leftarrow \arctan[\frac{\text{Im}(\boldsymbol{z})}{\text{Re}(\boldsymbol{z})}]$
5:         $\boldsymbol{B} \leftarrow \exp(i\vartheta^*)\boldsymbol{B}$
6:     **end if**
7:     **return** $\|\boldsymbol{A} - \boldsymbol{B}\|_\diamond$
8: **end function**

---

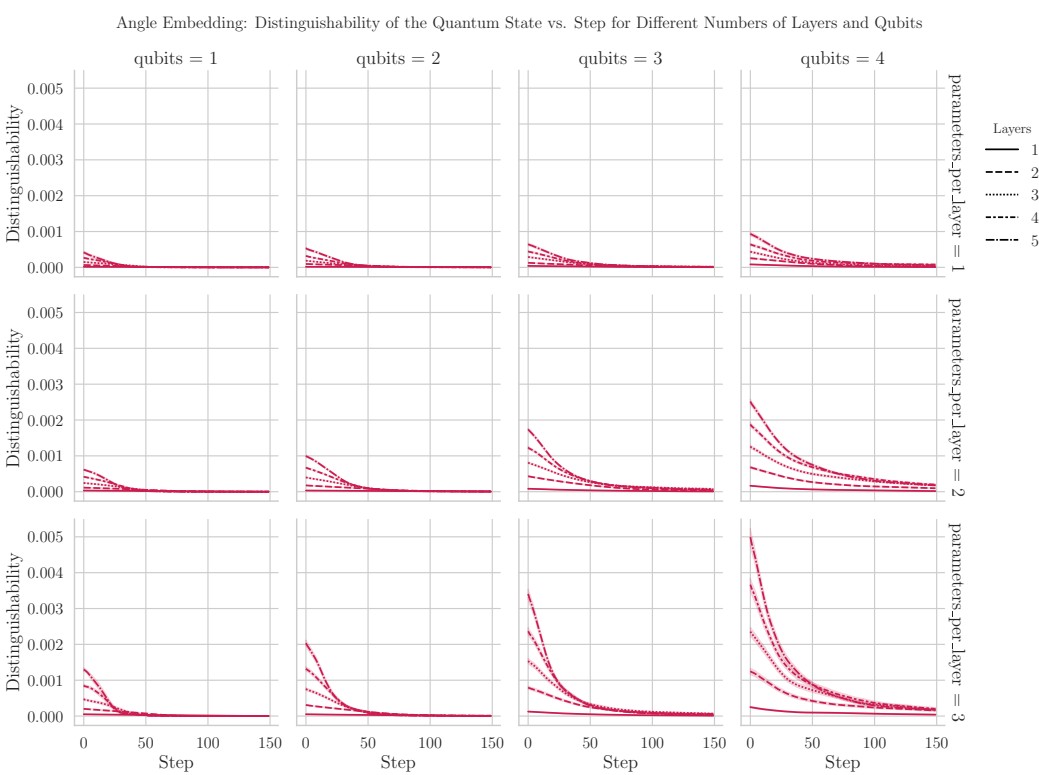

Figure 5: Channel sensitivity for angle embedding

