# OpenReview forum: "Exploring channel distinguishability in local neighborhoods of the model space in quantum neural networks"
_ICLR.cc/2025/Conference — ICLR 2025 Poster_

### Official Review · Reviewer_E1y9 · 2024-10-24

**Soundness:** 4
**Presentation:** 4
**Contribution:** 3
**Rating:** 6
**Confidence:** 5

**Summary:**

The authors introduce an "expressivity" measure of quantum neural network (QNN) architectures that they argue is more natural than those previously explored in the literature. One example of the latter is a difference measure between the given QNN architecture when randomly initialized and a $t$-design (for some small $t$); the authors show that having a large expressivity according to such a measure does not necessarily indicate that the model explores a large subspace of Hilbert space. This motivates their introduced definition, which is a sensitivity measure in diamond norm of the quantum network upon perturbing the network parameters. The authors then perform some numerical experiments indeed demonstrating that this measure tracks with how one would intuitively expect "expressivity" to behave (it grows with the number of layers and trainable parameters per layer, and so on).

**Strengths:**

Understanding measures of expressivity for QNNs is important for understanding the to-date loosely-defined "expressivity-trainability trade-off" known to exist in quantum networks (see, e.g., arXiv:2312.09121 and arXiv:2406.07072). The authors, in a very clear fashion, describe shortcomings of certain previously-introduced measures of quantum network expressivity; given these shortcomings, the authors then introduce a novel, natural notion of expressivity that circumvents these shortcomings.

**Weaknesses:**

Though not the fault of the authors, the introduced measure of expressivity has the unfortunate side-effect of being difficult to estimate; this limits the scope of their numerical experiments. It also makes the relative advantage of the authors' introduced expressivity measure over an expressivity measure based on closeness to the Haar measure (rather than a small-$t$ $t$-design) unclear, given both are difficult to estimate.

**Questions:**

Is there a natural reduction of certain, previously-explored measures of expressivity to the authors' introduced measure of expressivity? For instance, I believe demonstrating that having large expressivity based on a measure of closeness to the Haar measure, and showing this implies a large expressivity according to the authors' definition (averaged over parameter space), would strengthen this work.

Furthermore, what relative advantages do the authors' introduced expressivity measure have over expressivity measures based on closeness to the full (i.e., not $t$-design) Haar measure? This impacts the broader reach of this work, as to me it is currently unclear what the relative merits are of the introduced measure to this previously-studied measure.

---

> ### Author Response · Authors · 2024-11-20
>
> Firstly, we would like to thank the reviewer for the comments and questions.
>
> We acknowledge that we did not sufficiently explain the term expressivity and our measure as a comparison. In particular, the following (or adaptations thereof) will be added to the rebuttal.
>
> Ansatz Expressivity: "We acknowledge a missing clear definition of expressivity in QML for now. It is sensible, therefore, to transfer it from classical ML, where model expressivity is defined as the range of functions a model is able to compute (arXiv:1606.05336). Hence, it is a function of architecture $A$, input $x$ and parameters ${\vartheta}$; $F_A(x;{\vartheta})$. On the contrary, our work lies in the premise of data-agnostic characterization of models. To this end, we will only consider \textit{ansatz expressivity}; the set of functions that can be generated solely by an ansatz itself."
>
> Issues with definition: "If one were to define expressivity as the capability of a model to represent all possible sets of unitaries, it would be required to go way beyond second moments (for, e.g., Kurtosis, off-diagonal correlations etc.) to ensure the necessary weights are captured where there is sufficiently dense population. As the moment operator requires tensor products proportional to the moments, this becomes computationally infeasible very quickly. It is not clear how to overcome the curse of dimensionality while at the same time proposing a measure that captures expressivity adequately; in fact, most tools from QIT suffer from this very phenomenon, which speaks to the difficulty of designing such a metric. Further, there already exists a line of non-unitary VQA research (e.g., arXiv:1810.03787, arXiv:2411.05760), that needs to be considered as well. As expressivity is currently the most common metric used to describe the ansatzes, our work employs an alternative approach to analyze the model space, *unrelated to expressivity*, but highly relevant for trainability and training dynamics. Therefore, we focus on local neighborhoods of the model space, in particular, we study how the QNN changes upon perturbation of parameters. This allows analyzing training dynamics and tracking the extent to which the QNN changes during training."
>
> Therefore, we do not aim to come up with another measure of expressivity (as ours only measures distances in the unitaries in a local neighborhood), as the term expressivity is inherently undefined. It could be defined w.r.t the full Haar measure (would ensure density), however, this is not a tangible measure, and, would not meaningfully capture expressivity for other architectures, such as non-unitary architectures (e.g., arXiv:2312.09121 and arXiv:2406.07072).
>
> Our aim is to provide useful measures to understand the models themselves (without data), and the most widely used measure is expressivity w.r.t. 2nd moments of the Haar measure (cf. paragraph on p.5 "Currently, ...").
>
> With that in mind, we would love to elaborate on the raised questions:
>
> 1. It is computationally infeasible to compare to the full Haar measure (indeed the tensor products involved in the moment operator restrict its usage to small k-designs even for computers with large memory). The computational intractability can be tied to the fact that Haar-random unitary requires a depth exponential in the number of qubits (quantum register size). Thus, any efficient construction of random unitaries requires a notion of approximation (truncation of the full Haar measure up to the t-th moment), which is discussed in detail in the seminal works (https://arxiv.org/abs/quant-ph/0611002v2, https://doi.org/10.1103/PhysRevA.80.012304). Due to the aforementioned engineering bottlenecks, no such work exists along these lines to the best of our knowledge. Further, the aim of our work is to study local neighborhoods of the model space, hence, it is not aimed to average over the parameter space.
>
> 2. The aim of the work is to showcase that training difficulties for HEAs may very well be due to the architectures themselves, as the unitary that is applied is not distinguishable from itself upon parameter update. This framework could be transported to other (future) architectures.

---

> > ### Comment · Reviewer_E1y9 · 2024-11-20
> >
> > We thank the authors for their changes to the manuscript, which aids in its readability.
> >
> > To clarify my point on the Haar measure (my first question/what I perceive to be the weakness of the work): I agree with the authors that the complexity of calculating the expressivity with respect to truly Haar-random unitaries is exponential in the number of qubits. However, my concern is that calculating the authors' introduced diamond norm measure _also_ takes time exponential in the number of qubits. This echoes one of the concerns of Reviewer U5bz. To state my concern as a question: how does the complexity of estimating the authors' proposed expressivity measure scale compared to that of using the Haar measure?

---

> > > ### Author Response · Authors · 2024-11-25
> > >
> > > We thank the reviewer for this clarification.
> > >
> > > The reviewer is absolutely right in that the calculation of the diamond norm scales badly for system sizes (as the Haar measure does). As a result, we derived the clear dependence on the number of parameters for HEAs. Our aim is by no means to argue to calculate the diamond norm for all architectures one would like to train; rather, (generally) it should open a new perspective on the model space and (for HEAs) give more insights from the ansatz perspective on why such architectures may not be useful on small scales. QIT tools are very useful in that they have an interpretation and intuitive meaning attached to them, which we exploit for our definition of channel sensitivity. However, most of them are hard to compute in practice. Therefore, bounding such quantities is useful as it keeps the interpretative meaning while (mostly) yielding something computable.
> > >
> > > Future research on different architectures could aim to upper bound the associated channel sensitivity by something computable and (hopefully) come up with architectures that are, with more likelihood, distinguishable upon parameter update. We would like to stress, however, that it is not our aim to give an alternative computable metric with the channel sensitivity to the expressivity measure stated; the aim is to give a different perspective on the model space as the *concept of expressivity is inherently undefined*.

---

### Official Review · Reviewer_kPLW · 2024-10-28

**Soundness:** 3
**Presentation:** 4
**Contribution:** 3
**Rating:** 6
**Confidence:** 2

**Summary:**

The paper studies quantum neural network expressivity and model distinguishability. The aim is to better understand, at least for small architectures, the model space of so-called Hardware Efficient Ansatzes (HEAs) for quantum neural networks. To do this, the authors first argue why 2-design is an inadequate measure for model expressivity.  Then they define channel sensitivity to measure the effect of parameter changes on quantum neural network output using the diamond-norm from the literature. A Taylor-expansion-based argument allows to provide a natural upper bound on channel sensitivity. Moreover, the paper conducts an extensive numerical study to experimentally measure channel sensitivity for standard HAE quantum neural networks. As its main message, the paper states that its results ultimately suggest that iterative training of small quantum models may not be effective.

**Strengths:**

Overall, the proposed research question appears relevant, as a theoretical understanding of quantum neural networks is, to a large extent, still missing. The paper is mostly clearly structured and well-presented. Although limited to small architectures, the considered setting is still of interest, given the hardware limitations. Numerical study considers several important combinations of architectures.

**Weaknesses:**

Some of the key conclusions drawn from the numerical results are only partially justified by what is presented in the paper, weakening the overall significance of the reported results. More specifically:

-	Some important details regarding the setup of numerical experiments (employed optimizer, optimizer hyperparameters) and results (confidence level for confidence intervals) are currently missing in the paper. Also, it appears that robustness with respect to choice of optimizer / hyperparameters has not been studied.
-	The upper bound on channel sensitivity is an upper bound, not an equality. Some of the conclusions drawn in the discussion/abstract/introduction would only be valid in case of equality. For example, in the Discussion section, the paper states that channel sensitivity "scales with (1) the magnitude of changes and (2) the number of parameters." This statement is true for the upper bound, but not necessarily for the channel sensitivity, as the upper bound is very loose (as the experiments show) and no lower bound seems to be available. Similarly, the introduction the authors write as one of their four main contributions: "we introduce a measure for the distinguishability of an ansatz upon parameter update and provide an upper bound. It shows that many parameters are required to be able to distinguish the two operations, contrasting with the initial framework." This would be true in case of an equality, but an upper bound (which is loose, as the experiments show) does not seem sufficient to support such a claim.
-	As one of their four main contributions, the authors state "We provide numerical evidence that each ansatz is hardly distinguishable from itself". This seems to be based on Figure 3. Since Figure 3 also shows the (loose) upper bounds, the different scale of bounds and channel sensitivities in Figure 3 makes it impossible to see this solely from the figure.
-	Terminology is slightly confusing:  in Holmes et al (2021) the term expressibility was used. For classical neural networks, expressivity refers to their capability of approximating arbitrary given unknown functions. This is a fundamental property for choosing which neural network architecture to use, but does not seem to be discussed in the paper.

**Questions:**

1.	Which optimizer (and optimizer hyperparameters) did you use in the experimental results?
2.	How do the empirical results depend on the choice of optimizer and its hyperparameters?
3.	Where does the factor $\frac{1}{2}$ in (30) come from? Inserting (25) into (29) would give (30), but without additional factor $\frac{1}{2}$.
4.	Why do you use mean-squared loss, even though you are solving a classification problem?
5.	From its definition, the diamond norm seems to depend on number of qubits. How does an increase in number of qubits affect the diamond norm?
6.	Can you provide an additional figure, which is the same as Figure 3 but without the upper bounds?
7.	The reported results are averaged over architectures with the same number of parameters. Can you be more specific about what type of "confidence intervals" you show in the plots?
8.	For comparison reasons, it would be very insightful to analyse channel sensitivity of a non-entangled circuit. For example, for a circuit applying an $R_X$ gate to each qubit independently, what channel sensitivity do you get for $1,2,3,4$ qubits?
9. How do the results depend on your choice of input-encoding?

---

> ### Author Response · Authors · 2024-11-20
> **Part 1**
>
> The authors would like to thank the reviewer for the extensive comments and questions. We are happy to elaborate on them in the following.
>
> Ad Weaknesses:
>
> 1. Thank you for pointing this out, we forgot to mention it and will add the setup in the rebuttal version. For the second part of the comment, we would like to add that the experimental results we conducted merely support our theoretical bounds (which are independent of optimizer as the Taylor expansion corroborates with a mathematical framework this behaviour analytically). The aim of our paper is not to conduct an extensive benchmarking, rather, to show that HEAs suffer from these issues and to give numerical evidence beyond the theory.
>
> 2. We acknowledge that it is very important to distinguish between what follows from the bound compared to the experiments, however, we believe we drew the distinction clearly. Therefore, in the discussion section we do not state that the (empirical) channel sensitivity scales with the magnitude of changes and number of parameters, rather we say the "maximal distinguishability of quantum states" (which follows from the analytic upper bound). In particular, in the perturbative scheme, the upper bound that we provide shows that the maximum distinguishability could be at most $\sum_i\delta_i/2$, which is again confirmed in the experiments. The same holds for the last point. Our bound shows that many parameters are needed to distinguish them, however, the fact that the channel sensitivity is much smaller experimentally only \textit{strengthens} this point, because one would need many more parameters (which is also in line with the scaling in empirical channel sensitivity in terms of qubits and parameters, cf. Fig 4 (in initial version) and Fig 3 (in rebuttal version)). We would be grateful if the reviewer could provide us with a clarification on that issue.
>
> 3. We acknowledge this in the paper, therefore, we provided Figure 4. We will however move Figure 3 as it is currently to the appendix and replace it with the same figure without the bounds, as requested by Reviewer 2.
>
> 4. Thank you for pointing this out; we were indeed not specific enough with the definition of expressivity. We also actively chose to stay with the word "expressivity" instead of "expressibility" to advocate for transferring concepts that exist in classical ML to QML. We will add the following explanation to the rebuttal version. Further, regarding the second point, it is acknowledged in classical literature (arXiv:1606.05336) that this property is incredibly hard to capture, thus, it is not a property that is generally considered to choose a specific architecture (being a universal approximator in the limit might, but not expressivity per-se). We will also add an additional paragraph to the rebuttal version arguing why we chose to not continue with expressivity (and rather come up with a figure of merit).
>
> Ansatz Expressivity: "We acknowledge a missing clear definition of expressivity in QML for now. It is sensible, therefore, to transfer it from classical ML, where model expressivity is defined as the range of functions a model is able to compute (arXiv:1606.05336). Hence, it is a function of architecture $A$, input $x$ and parameters ${\vartheta}$; $F_A(x;{\vartheta})$. On the contrary, our work lies in the premise of data-agnostic characterization of models. To this end, we will only consider \textit{ansatz expressivity}; the set of functions that can be generated solely by an ansatz itself."

---

> > ### Author Response · Authors · 2024-11-20
> > **Part 2**
> >
> > Ad Questions:
> >
> > 1. Thank you for pointing this out. We will add the following information: Adam optimizer with step size 0.1, betas 0.9 and 0.99, epsilon 1e-08. It is not possible to predict what would happen with different optimizer and hyperparameter, however, as long as the step size is small enough to justify the Taylor expansion (which is perfectly in line with what is expected in training) it will be according to the bounds. Still, the aim of our experiments is to support our bounds and question the role of HEAs as an architecture, not to conduct an extensive benchmarking.
> >
> > 2. The prefactor 1/2 follows from the definition of the rotation gates $R_X(\theta) = e^{-i\theta\sigma_x/2}$. The half factor features in the Pauli rotation gates or alternatively in the Lie-algebra generators.
> >
> > 3. We do binary classification, therefore it is fair to use MSE. This is largely in line with literature (generally the expectation value is in [-1,1], so any entropy-based losses do not arise that intuitively).
> >
> > 4. This is implicitly covered in how we define the architecture. Generally, the diamond norm grows with the number of qubits, but we established a direct correspondence with the parameters instead (which will for generic architectures scale with the qubits in any way).
> >
> > 5. We will change this in the next version, thank you for pointing that out!
> >
> > 6. Thank you for pointing this out; we indeed did not mention it. The confidence intervals are on a 95\% confidence level, which will be added to the paper.
> >
> > 7. We would appreciate a bit more of a clarification for this question. It seems this could be interesting from a QIT perspective, however, it is not relevant for the ML use case. If all qubits (=features) are supposed to be causally related to the output, they need to be in the reverse lightcone (we measure on the first qubit). Therefore, increasing the number of qubits when we do not entangle the architecture will not change the ML model, as in any case only the first qubit contributes. If this does not answer the question, we would greatly appreciate some more details.
> >
> > 8. The bound does not (due to unitary invariance), however, the updates during the experiments might be done differently. The goal is, however, to get a measure that is independent of the input state (hence, data + (non-trainable) feature encoding). As requested by Reviewer 2 as well, we reran the experiments with angle embedding (the plots are added to the anonymized github: https://anonymous.4open.science/r/distinguishability-qnn-5C41/diamond-norm_angle.pdf, https://anonymous.4open.science/r/distinguishability-qnn-5C41/diamond-norm_amplitude.pdf). As expected, the embedding does not impact the channel sensitivity a lot. Unless requested by the reviewer, we would, however, like to not include this in the final paper as (1) experiments are not the main contribution, rather supporting our theory (also feature encoding is per unitary invariance not relevant for the theory), (2) the additional experiments do not contribute much more information as they again show that the distinguishability is small (the slightly different curves are negligible considering that distinguishability is maximum 0.5\% in both encodings; further, deviations in the $10^{-3}$ should be analyzed with caution considering numerical instabilities of linear algebra packages (SVD within diamond norm)), (3) we are already at the maximum page limit (generally, rather section 3 and 4 should get more space). If requested, we could add it to the appendix, but we feel the added information is minimal and the appendix is quite extensive at this point. We would be grateful for a discussion and clarification.

---

> ### Comment · Reviewer_kPLW · 2024-11-22
>
> I would like to thank the authors for the clarifying comments, which have addressed several of my concerns. I have adjusted the ratings accordingly.
> Regarding my comment that robustness with respect to choice of optimizer / hyperparameters has not been studied, the authors state: "For the second part of the comment, we would like to add that the experimental results we conducted merely support our theoretical bounds". On the other hand, the upper bound on channel sensitivity is very loose - as the graphs in the original submission showed - so that its usefulness remains unclear.
>
> On a side note, maybe it might be helpful for the readers to resolve the notational inconsistency between (25)-(26) and (29)-(30), and introduce the factor 1/2 also the previous section.

---

> ### Author Response · Authors · 2024-11-25
>
> Thank you for clarifying the comment, the concern is now quite clear to us. We acknowledge the imprecision in the original paper  stating that the bounds are loose, this will be corrected in the rebuttal version. It can generally not be said that this is the case, as the bound could very well be achieved at some point in the model space (e.g., consider variance in Fig.2). The only thing that our experiments show is that it is unlikely to be the case in large parts of the model space.
>
> As we argue, the main aim of the paper is to show that HEAs have architectural issues on small scales. As discussed with Reviewer 4 (E1y9), the quantity is fundamentally difficult to calculate due to the computational complexity of the diamond norm. Rather, it should be used as a figure of merit that could be upper bounded by future architectures to keep the nice interpretation while giving something computable.
>
> As discussed with Reviewer 2 (U5bz), there is absolutely no use of it during actual training runs, as one might just train the model. The channel sensitivity remains a figure of merit to assess the model space and associated training dynamics. The quantity is *not* influenced by the optimizer used, only the *magnitude of updates* are (which are small, due to the framework of VQAs). As to the point of the usefulness being unclear, we would also like to draw the reviewer's attention to the discussion with Reviewer 2 (U5bz) about this quantity being not trivially small for every architecture, rather, the very definition of a HEA allows the dependence on the partial derivatives of the unitary in the bound to vanish.
>
> Thank you for pointing us towards the notational inconsistency. Equations (25), (26) will be changed accordingly.

---

### Official Review · Reviewer_U5bz · 2024-10-31

**Soundness:** 4
**Presentation:** 3
**Contribution:** 3
**Rating:** 8
**Confidence:** 3

**Summary:**

The paper considers methods of characterising the variational ansätze represented within quantum neural networks. The paper argues against the use of so-called 2-designs as a measure for expressivity that have been used in previous works. The work goes on to introduce a notion of channel sensitivity, which considers how much (in terms of a so-called diamond norm, a norm used more in Quantum Information theory) a unitary changes under perturbation of variational parameters. Bounds for the channel sensitivity are derived by consideration of Taylor series expansions and used to argue against the distinguishability of ansätze with few variational parameters on parameter perturbation. Numerical studies then investigate the notions introduced above in terms of both random perturbations of variatonal parameters, and also perturbations introduced during a training loop.

**Strengths:**

1. The introduction and background material are particularly well-written and offer a broader ICLR readership a succinct way of learning about quantum neural networks. However, it is possible that too much space has been allocated within this area at the expense of a fuller exposition of novel aspects of the work.
2. The work takes aim at foundational issues in variational quantum circuits and the points made are widely applicable to QNNs as a whole.
3. The proposed channel sensitivity metric based on diamond norms seems to be aiming at something principled, in that it seeks to characterise the action of a unitary on any state. Its application to variational quantum circuits, to my knowledge, is novel.

**Weaknesses:**

1. Sections 3 and 4 feel like they should be the main contributions of the paper, but they squeezed into about a single page. For example, precise mathematical definitions of the how the quantum states in Eq (4) are missing from the main text, which makes it hard to follow.
2. The work as a whole considers expressivity of a variational quantum circuit as a problem of the ansatz and does not give any weight to the importance of feature encoding into the quantum circuit. It is known that the spectral representation of a QNN is directly given by the feature encodings ([Schuld, 2021](https://arxiv.org/abs/2008.08605)).
3. The diamond norm under parameter change seems like an interesting metric, but it remains unclear what can be done with it.
4. Similarly, the bound on the channel distinguishability under small perturbations seems not to be useful. It seems natural that the QNNs will only change gradually under small parameter perturbations such as those encountered during training, so what use is an upper bound?
5. The main points of the work seem somewhat incongruous: the criticism of the 2-design as an expressivity metric seems like it should give way to an alternative expressivity metric. However, the channel distinguishability metric proposed does not provide an expressivity metric.

**Questions:**

Question:

1. My understanding is that the diamond norm is derived to provide a channel distinguishability based on a single measurement of the state. Does this translate to distinguishability of a QNN model, where i) we might make use of several measurements and ii) one might make use of a specific measureable (e.g. single qubit magnetisation) rather than the whole state?
2. What can be done with the proposed channel distinguishability metric? The fact that it can be bounded from above for small perturbations does not seem useful. All neural networks (classical or quantum) typically won't change too much in a single training step---why is upper bounding this change in terms of some norm useful to a practitioner?

Suggestions:

2. I think section 3 and 4 can be elaborated on more. In particular, the criticism of 2-designs as a measure of expressivity is hard to read without going back and forth between the main text and the appendix several times.
3. Elaborating on section 3 and 4 would probably need more space. Some of the figures can be moved to the appendix, e.g. the three rows of figure 2 are conveying a similar message, so perhaps two of the rows can be placed in the appendix? Same with figure 3. The conclusions and discussion could be shortened as well.
4.  Ensure that all figures are included in a vector format or lossless format. Maybe I'm not seeing things right, but the quality of the figures seems a little bit off.
5. A minor point unrelated to the core contributions of the paper. But amplitude encoding of features into a QNN feels like a non-standard way of conducting the numerical experiments. Apart from amplitude encoding not being near-term friendly, the resulting QNN will not be particularly expressive. I.e. one will end up with something like `<PCA|Hermitian|PCA>`, which, since the PCA is a linear combination of features, will result in a sum of pairwise interactions with no zeroth, first or cubic and higher order terms. Encoding PCA components into single-qubit rotations would be a more standard approach, to my knowledge.

---

> ### Author Response · Authors · 2024-11-20
> **Part 1**
>
> We want to thank the reviewer for the comments and very valuable suggestions.
>
> Concerning the comment toward the weight of the feature encoding in expressivity. While our paper only focuses on ansatz characterization/expressivity (which we state in the abstract), the initial submission failed to convey this message in the main parts. We will add the following to the final version, clarifying terminology better.
>
> Ansatz Expressivity: "We acknowledge a missing clear definition of expressivity in QML for now. It is sensible, therefore, to transfer it from classical ML, where model expressivity is defined as the range of functions a model is able to compute (arXiv:1606.05336). Hence, it is a function of architecture $A$, input $x$ and parameters ${\vartheta}$; $F_A(x;{\vartheta})$. On the contrary, our work lies in the premise of data-agnostic characterization of models. To this end, we will only consider *ansatz expressivity*; the set of functions that can be generated solely by an ansatz itself."
>
> Issues with definition: "If one were to define expressivity as the capability of a model to represent all possible set of unitaries, it would be required to go way beyond second moments (for, e.g., Kurtosis, off-diagonal correlations etc.) to ensure the necessary weights are captured where there is sufficiently dense population. As the moment operator requires tensor products proportional to the moments, this becomes computationally infeasible very quickly. It is not clear how to overcome the curse of dimensionality while at the same time proposing a measure that captures expressivity adequately; in fact, most tools from QIT suffer from this very phenomenon, which speaks to the difficulty of designing such a metric. Further, there already exists a line of non-unitary VQA research (e.g., arXiv:1810.03787, arXiv:2411.05760), that needs to be considered as well. As expressivity is currently the most common metric used to describe the ansatzes, our work employs an alternative approach to analyze the model space, *unrelated to expressivity*, but highly relevant for trainability and training dynamics. Therefore, we focus on local neighborhoods of the model space, in particular, we study how the QNN changes upon perturbation of parameters. This allows analyzing training dynamics and tracking the extent to which the QNN changes during training."
>
> Regarding the last comment; it is also covered by the sections outlined above that will be added to the paper. The fact that it is not clear what a maximally expressive QML model is (full Haar measure might be an (uncomputable) measure, but would not cover emerging non-unitary models), makes it incredibly hard to define an alternative metric. Therefore, we propose to look at local regions of the model space to come up with meaningful and useful properties of ansatzes; we hope this clarification makes our motivation clearer.
>
> Questions:
>
> 1. We would be grateful for a clarification of (i), as we are not entirely sure what is meant. If the reviewer refers to several measurements in terms of shots and getting an expectation value, then, generally no, because the diamond norm is defined according to the single use distinguishability, but there are implications for the expectation value based on the single-use case (if we have a 5\% chance of distinguishing them in a single use, this has implications for how often the result will be different when averaging over 100s of shots). The diamond norm aims to upper bound on any measurement, thus answering (ii).
>
> 2. While it is true that (Q)NNs are supposed to only change gradually under parameter update, it is expected that the unitary transformation changes and is distinguishable. Intuitively, if parameters are updated but the unitary is only with very small likelihood distinguishable from the previous function (e.g., one would not even get a different result), it is incredibly difficult to train. Therefore, upper bounding this metric is exactly what we required to show that HEAs are fundamentally difficult to train on small scale. The measure therefore is not necessarily only useful to a practitioner, but rather we envision that it can be used as a guiding point for future architectures (beyond HEAs) to evaluate and bound training dynamics.

---

> > ### Author Response · Authors · 2024-11-20
> > **Part 2**
> >
> > Suggestions:
> >
> > 1/2/3. We would like to thank the reviewer for their comments. We will rearrange the figures in the rebuttal version and restructure section 6; in particular, move the bound to section 4. Indeed, it is more fitting there, especially considering that this is our main contribution. Further, the figures are indeed a bit off, we will correct that for the rebuttal version.
> >
> > 4. Thank you for pointing this out. Generally, to the best of our knowledge, there are three commonly used methods, amplitude, angle (which you mentioned), and IQP embedding, however, no clear standard has crystallized (probably partially because there are hardly any works studying their effect on the model). Nonetheless, we decided to rerun the experiments using angle embedding and add the plots to the anonymized github (https://anonymous.4open.science/r/distinguishability-qnn-5C41/diamond-norm_angle.pdf, https://anonymous.4open.science/r/distinguishability-qnn-5C41/diamond-norm_amplitude.pdf). As expected, the embedding does not impact the channel sensitivity a lot. Unless requested by the reviewer, we would, however, like to not include this in the final paper as (1) experiments are not the main contribution, rather supporting our theory (also feature encoding is per unitary invariance not relevant for the theory), (2) the additional experiments do not contribute much more information as they again show that the distinguishability is small (the slightly different curves are negligible considering that distinguishability is maximum 0.5\% in both encodings; further, deviations in the $10^{-3}$ should be analyzed with caution considering numerical instabilities of linear algebra packages (SVD within diamond norm)), (3) we are already at the maximum page limit (as pointed out in suggestion 2, rather section 3 and 4 should get more space). If requested, we could add it to the appendix, but we feel the added information is minimal and the appendix is quite extensive at this point. We would be grateful for a discussion and clarification.

---

> > ### Comment · Reviewer_U5bz · 2024-11-20
> >
> > The proposed amendments to the paper sound like they'd improve the exposition greatly.
> >
> > However, my principal concern remains regarding i) the utility of the diamond norm ii) conclusions regarding the difficulty of training of HEAs, which I do not believe are adequately supported.
> >
> > _On concerns of conclusions applying to HEA_
> >
> > Since parameter updates are small for gradient-based optimisation, it seems only natural that the channel distinguishability will be small for _every_ ansatz during successive parameter updates, not just HEA.
> >
> > In classical ML as well, neural networks (both the weights and the outputs) don't normally change much in a single training step. That's not to say that they aren't trainable: we're able to identify parameter gradients. How is this case with HEA any different? It seems natural that the channel-distinguishability should be small on single parameter updates, but we can still calculate gradients of HEA parameters and take many successive changes.
> >
> > _On the practical utility of channel distinguishability_
> >
> > The theoretical upper bounds seem hard to put into practice: i) they are predicated on the changes being small in the first place, which seems to directly imply small channel distinguishability anyway and ii) they need to be evaluated locally in parameter space, which doesn't give a wholistic view of an ansatz.
> >
> > Consequently, the main utility seems to be the experimental calculation of the channel distinguishability. But channel distinguishability is expensive to calculate for even small circuits, so it seems hard to use empirically as well. If channel distinguishability needs to be empirically determined with parameter update directions given by optimisation on a dataset, is it not easier just to evaluate the QNN on said dataset before and after parameter updates?

---

> > > ### Author Response · Authors · 2024-11-25
> > > **Part 1**
> > >
> > > Thank you for the clarification, we can better understand the concerns now. It is, however, non-trivial that the changes in unitary are small, which we will prove in the following. While *parameter updates* are small in gradient-based optimization, the changes in unitary are highly dependent on the architecture of the circuit. In particular, we derive the following in the paper.
> > >
> > > $\|U(\vartheta) - U(\vartheta + \delta)\|_\Diamond $
> > >
> > > $\approx \|U(\vartheta) - (U(\vartheta) + \sum_j \delta_j\frac{\partial U(\vartheta)}{\partial\theta_j})\|_\Diamond$
> > >
> > > $ = -\|\sum_j \delta_j\frac{\partial U({\vartheta})}{\partial\theta_j}\|_\Diamond$
> > >
> > > Due to homogeneity and the triangle inequality, we get the following inequality (the $\sim$ follows from truncation of $O(\delta^2)$).
> > >
> > > $\|U({\vartheta}) - U({\vartheta} + {\delta})\|_\Diamond $
> > >
> > > $ \lesssim \sum_j|\delta_j|\|\frac{\partial U({\vartheta})}{\partial\theta_j}\|_\Diamond$
> > >
> > > Therefore, it follows that the upper bound is dependent on both the perturbation $\delta$, and the sum of partial derivatives of $U$. The fact that these factors cancel is dependent on the structure of the Jacobian matrix of $U$, i.e., partial derivatives of U w.r.t. to the unperturbed parameters, for HEAs. Therefore, the very fact that we consider HEAs (and the property that their generators are also unitary) is the fact that they cancel. The following shows, that the partial derivatives are non CPTP if the generators are nonunitary, hence, the individual diamond norms will not be upper bounded by 2. In the following, we consider trainable unitary gates with non-unitary, fixed, Hermitian generators (i.e., $H_j^2 \ne \mathbb{I}$).
> > >
> > > A self-similarity transformed density operator $\rho$ is defined as,
> > >
> > > $\mathcal{E}(\rho) := V(\vartheta) \rho V(\vartheta)^{\dagger}$
> > >
> > > The map above is said to be a valid quantum channel, if it is a completely positive trace preserving (CPTP) map.
> > >
> > > **Proposition 1:** The partial derivative of a unitary gate  w.r.t a parameter, i.e., $\frac{\partial V(\vartheta)}{\partial \theta}$ is non-unitary if the generator is non-unitary.
> > >
> > > **Proof:** Let $V(\vartheta) = \exp\{(-i \frac{\vartheta}{2} H_j)\}$  be a unitary quantum gate, where $H_j$ is a non-unitary generator (fixed, $H_j^2 \ne \mathbb{I}$).  It is easy to verify that $\frac{\partial V}{\partial \theta} = - i \frac{H_j}{2} V(\vartheta)$ does not satisfy the unitarity condition, i.e. $(\frac{\partial V}{\partial \theta})^{\dagger} \frac{\partial V}{\partial \theta} \neq \mathbb{I}$.
> > >
> > > **Proposition 2:** The partial derivative of a unitary gate w.r.t the parameter $\vartheta$ is not a trace-preserving map, if the generator of the gate is non-unitary.
> > >
> > > **Proof:** It is easy to observe that $\text{Tr}\Big( (\frac{\partial V}{\partial \theta}) \rho (\frac{\partial V}{\partial \theta})^{\dagger}\Big) = \text{Tr}\Big(-\frac{H_j}{2} V(\vartheta) \rho \frac{H_j}{2} V(\vartheta)^{\dagger}\Big)$. Using the invariance of the trace under circular shifts, one gets $\text{Tr}\Big( (\frac{\partial V}{\partial \theta})\rho (\frac{\partial V}{\partial \theta})^{\dagger} \Big) = \frac{1}{4} \mathbb{I}$, hence proves that it does not preserve the trace, if $H_j$ is nonunitary.

---

> > > > ### Author Response · Authors · 2024-11-25
> > > > **Part 2**
> > > >
> > > > **Proposition 3:** The partial derivative of a unitary gate w.r.t the parameter $\vartheta$ is not a completely positive map if the generator is non-unitary.
> > > >
> > > > **Proof:** To prove this, we could use one of the three equivalent conditions of the Choi's theorem on completely positive maps (https://doi.org/10.1016/0024-3795(75)90075-0). We prove this using the extended system argument following Choi's theorem.
> > > >
> > > > A linear map acting between vector spaces of linear operators (so-called *superoperator*) $\mathcal{E}: \mathbb{C}^{n \times n} \rightarrow \mathbb{C}^{m \times m}$ is completely positive, if it maps any positive semidefinite operator to another positive semidefinite operator. Although, this condition holds for any positive semidefinite operator,  we will explicitly prove this for the density matrix $\rho$, since every density matrix is positive semidefinite (i.e., $\sum_i p_i |\langle\phi|\psi_i\rangle|^2 \geq 0$).
> > > >
> > > > The larger system is defined as:
> > > > $(\mathbb{I}_n \otimes \mathcal{E})(\rho) \in \mathbb{C}^{n \times n} \otimes \mathbb{C}^{m \times m}$
> > > >
> > > > Substituting the action of the linear map from the superoperator definition into the above for the gradient of the unitary gate $\mathcal{E} =\Big( (\mathbb{I}_n \otimes \frac{\partial V(\vartheta)}{\partial \vartheta}) \rho (\mathbb{I}_n \otimes\frac{\partial V(\vartheta)}{\partial \vartheta})^{\dagger} \Big)$. Completely positive implies the above is positive, i.e. $ (\mathbb{I}_n \otimes \mathcal{E})(\rho)  \geq 0$. This operation fails in being completely positive, since we get factors of $-1$ coming from the $i^2$ factor due to derivatives. Thus, proving our proposition.
> > > >
> > > > **Proposition: Jacobian of a unitary matrix is not a CPTP map**
> > > >
> > > > Using propositions (1, 2, 3), this can be generalized for the multi-parameter (vector-valued) setting, for a Unitary dependent on set of parameters $\vartheta$ and in general a quantum circuit defined as:
> > > >
> > > > $U(\vartheta) := \prod_j W_j \exp \{(-i \frac{\vartheta}{2} H_j)\}$
> > > >
> > > > where, $W_j$ are set of fixed gates carrying no parameter dependence.
> > > >
> > > > **Argument :** Thus Non-CPTP maps (non admissible maps) generate unphysical outputs. This is because, due to its non-unitarity, and not complete positive properties, it does not map a density matrix to another valid density matrix.  Moreover, it is not bounded by 2 (in the case of differences of CPTP maps) and can exceed 2 due to matrices with arbitrarily large trace norms. The lack of trace preservation or complete positivity property removes the constraints that limit the norm. This is what typically occurs, wherein $||\frac{\partial U(\vartheta)}{\partial \vartheta}||_\Diamond$, and can be arbitrarily large owing to its non-CPTP (non-admissible maps) nature.
> > > >
> > > > Therefore, the fact that the partial derivatives evaluate to 1 is nontrivial and will not hold for *any* ansatz. Per se, any upper bound on this quantity depends not only on $U$, but also on its local structure around parameters $\vartheta$. Following our proof, this will not be trivially small due to not being CPTP. Should the reviewer have any references that support their statement that this is always small, we would highly appreciate clarification. Should the paper be accepted, we are happy to add this to the Appendix in the camera-ready version after further polishing and integration with the rest of the paper and related work.
> > > >
> > > > The issue we point out is that in small QNNs with HEA, the channels are *almost indistinguishable*. This does not mean that the parameters do not change slowly (which is a necessity in iterative/gradient-based training), but it means that the applied operation is not discriminable. The argument we are drawing is similar to barren plateaus, that if the operations are not distinguishable, so will the outputs and resulting loss function. This leads to untrainable models.
> > > >
> > > > Regarding the practical utility; the fact that the upper bound is predicated on the changes being small is not a limitation, rather it is a feature of the VQA framework (generally any iterative optimization algorithm). As argued before, this does not directly imply small channel distinguishability for any ansatz. The fact that VQAs are trained classically gives rise to the changes happening in parameter space, but our work tackles this very part of the algorithm by checking the impact of updating the parameters on the unitary, thus providing novel insights. The fact that we can decouple the upper bound on the channel sensitivity for HEAs from the position in parameter space is thus indeed an interesting insight.

---

> > > > > ### Author Response · Authors · 2024-11-25
> > > > > **Part 3**
> > > > >
> > > > > The channel sensitivity is by no means meant to be used to train models. In fact, we question the utility of HEA (which has been questioned before: arXiv:2211.01477) from a novel perspective. The experimental calculation corroborates our theory to show that HEAs are fundamentally flawed on small architectures. There is no use of the channel sensitivity during training; the reviewer is absolutely right to mention this; it is indeed a theoretical figure of merit to assess architectures.

---

> > > > > > ### Comment · Reviewer_U5bz · 2024-11-25
> > > > > >
> > > > > > I appreciate the authors' efforts in clarifying several aspects for me. However, my principal concerns persist: the channel sensitivity is too expensive to use practically, and I still feel the theoretical bounds in the paper should be small for all ansätze, not just HEA, so that the conclusions and criticism concerning HEA are not sufficiently justified to my mind.
> > > > > >
> > > > > > In concrete terms:
> > > > > >
> > > > > > _Why the first order $\delta$ terms of channel distinguishability should disappear for all ansätze._
> > > > > >
> > > > > > Without assuming any specific ansatz, and recalling the definition of the channel distinguishability used in the paper $cs_U(\theta, \delta) = ||U(\theta) - U(\theta + \delta)||_\diamond$.
> > > > > >
> > > > > > Instead of expanding the unitary in $\delta$ as in the paper, you can expand the channel distinguishability itself around $\delta=0$.
> > > > > >
> > > > > > $$cs_U(\theta, \delta) \approx cs_U(\theta, 0) + (\nabla_\delta cs_U (\theta, \delta)|_{\delta=0})^T.\delta + O(\delta^2)$$
> > > > > >
> > > > > > The zeroth-order term in $\delta$ is zero trivially. However, the channel distinguishability is not only zero when $\delta=0$, it is also _minimised_ at $\delta=0$, consequently the gradient is also zero. So all first order terms in $\delta$ do not appear in this Taylor expansion for any ansatz.
> > > > > >
> > > > > > Consequently, for all ansätze, you need to go to at least second order in $\delta$ to approximate the channel distinguishability.
> > > > > >
> > > > > > The situation is similar in classical ML where one looks at the derivatives of the KL-divergence under parameter perturbations and finds that one needs to go to the second order (Fisher-Information Matrix) to get meaningful non-zero terms.

---

> ### Comment · Reviewer_U5bz · 2024-11-27
>
> To elaborate further on the last remark, some doubts about the the correctness of several contributions of the paper :
>
> __Small distingushability is not a result of features of quantum models but a mathematical property of norms__
>
> For _any_ sufficiently smooth function of trainable parameters $f(\theta)$ a parameter update $\delta$ and a smooth norm $|| . ||$ we have that $|| f(\theta) - f(\theta + \delta)|| = O(\delta^2)$ on expansion around $\delta = 0$. This is a property stemming from $||0|| = 0$ and $(\nabla_\delta(||f(\theta) - f(\theta + \delta))||))|_{\delta=0}=0$, since norms are minimised around zero.
>
> This applies to any function, whether it's quantum or classical. Note that $f$ is not the model output, it can be any function dependent on the parameters.
>
> The channel distinguishability proposed by the authors is defined in such a setting. Three of the four key contributions of the paper (lines 64-73) use the flatness of the channel distinguishability as a criticism of quantum models. But there is nothing innate about quantum models that causes this flatness: the metric proposed by the authors is itself locally flat.
>
> Without this point being addressed, the paper's key messages can be seen as misleading.

---

> > ### Author Response · Authors · 2024-11-28
> > **Part 1**
> >
> > We would like to thank the referee for the active participation in the discussion, which led us to the root of the misunderstanding. The concern is now very clear to us. We acknowledge the proposed method of instead expanding the channel distinguishability $cs_U (\vartheta, \delta)$ itself. For a consistent Taylor expansion, however, the gradient would have to be taken w.r.t. $\theta$ instead of $\delta$.
> >
> > $cs_U(\vartheta, \delta) = cs_U(\vartheta, 0) + (\nabla_{\theta} cs_U(\vartheta, \delta)_{|\delta=0})\cdot \delta + \mathcal{O}(\delta^2).$
> >
> > We acknowledge this may be a typo of the referee, which is why we explain below why differentiability cannot be trivially assumed. We also want to mention, that we did consider this during our work; this is a valuable research direction pointed out by the referee; however, we refrained from pursuing this approach due to the following reasons:
> >
> > * The partial derivatives or gradients (jacobians) of a (super)operator or a quantum channel $A$ wrapped within diamond norms $||A||_\Diamond$ cannot be trivially assumed to be differentiable, especially near critical points. The derivatives are non-trivial to compute (even with modern day tools such as automatic-differentiation) and they are often not well-defined objects, especially when the operator contains poles on the complex plane or when it contains zeros of the matrix-valued functions in the denominator. Thus, the quantity we are evaluating (at the parametrized quantum circuit level) depends on its behavior near zeros (we shall expand that these are primarily associated with the singular values of the operators/quantum channels).
> >
> > **Example**: Considering one of the most utilized and simple norms in quantum information, i.e., the Frobenius norm (2-norm) of the same quantity, i.e., $A(\vartheta) = U(\vartheta) - U(\vartheta + \delta)$, where $\delta$ is the perturbation, is defined as:
> >
> > $||A(\vartheta)||_2 := \sqrt{\text{Tr}(U(\vartheta) - U(\vartheta + \delta))^{\dagger} (U(\vartheta) - U(\vartheta + \delta))} $
> >
> > $= \sqrt{\sum_{ij} |A_{ij}(\vartheta)|^2}$, the gradient (Jacobian) w.r.t $\vartheta$ reads:
> >
> > $\nabla_{\vartheta} ||A(\vartheta)||_2 = $
> >
> > $\frac{1}{2\|A(\vartheta)\|_2}*$
> >
> > $\nabla_{\vartheta}(|\sum_{ij} \|A_{ij}(\vartheta)\|^2)$
> >
> > (Please consider the above two lines as one, OpenReview does not render it in math mode if we have it as one line)
> >
> > The derivative is thus
> >
> > $\nabla_{\vartheta} ||A(\vartheta)||_2^{2} :=$
> >
> > $ \nabla_{\vartheta} \sum_{ij} |A_{ij}(\vartheta)|^2 = 2 \text{\ Re(\ Tr}(A^{\dagger}(\vartheta) \nabla_{\vartheta} A(\vartheta)))$
> >
> > . Thus, combining all the above, one finds that:
> >
> > $\nabla_{\vartheta} ||A(\vartheta)||_2^{2} :=$
> >
> > $ \frac{1}{||A(\vartheta)||_2}*$
> >
> > $ {\ Re(\ Tr}(A^{\dagger}(\vartheta) \nabla_{\vartheta} A(\vartheta)))$
> >
> > (Note again, the above two equations should be in one line, OpenReview does not render it correctly)
> >
> > From this example, it is clear that there is a $\frac{1}{||A(\vartheta)||_2}$ which can contribute to larger values if the denominator is relatively small (which is the case when $\delta \sim 0$).
> >
> > * Pertaining to our use case, which is the computation of the derivative of the Diamond norm, it is not always guaranteed that a well-defined derivative (Jacobian) exists everywhere when evaluating on the parameter space values.
> >
> > * Diamond norms by definition are related to the *Schatten*-1 norm $\text{Tr} \sqrt{A^{\dagger} A}$ and are constructed out of the singular values of the unitary matrices (https://doi.org/10.1007/s004400100186). The distribution of a set of singular values $\sigma_i(A)$ are not always differentiable, especially if $A$ is (a) rank-deficient as compared to the dimension of the associated vector space (in our case the Hilbert space $\mathbb{C}^d$), and (b) due to non-differentiability of singular values which arises from
> >     - nearly degenerate or degenerate eigenvalues and singular values,
> >     - Rank deficiencies in $A$,
> >     - Piecewise behavior (meaning not a differentiable manifold) of the singular value functions.
> >
> > Hence, our canonical choice was to expand the unitaries via a  series expansion in the perturbations $\delta$ instead of expanding the channel sensitivity itself, which contains  complicated Jacobian/multi-variable partial derivatives of the *Diamond norm* of a matrix.

---

> > > ### Author Response · Authors · 2024-11-28
> > > **Part 2**
> > >
> > > The aforementioned reasons also corroborate that the gradient of the diamond norm for a generic architecture evaluated at the origin does not strictly correspond to a stationary point or a local or global extrema of the diamond norm, and strictly depends on the behavior of singular values, especially around the neighborhood of these critical points $\delta \sim 0$ (gradient could also be ill-defined when evaluated on those points). Of course, if $\delta = 0$, then it basically corresponds to no change in the architecture. We want to actively highlight that is not what we are doing. From a QML perspective, we want to quantify the change in unitary associated with a parameter update. Not changing the parameters means we would not train at all, thus it is not an interesting quantity from a training or model space perspective. Thus, in our experiments we change the parameters 1\%, 0.5\% and 0.1\% respectively, or even evaluate during model training.
> > >
> > > While it does hold true for certain architectures (i.e., smooth function of trainable parameters $f(\vartheta)$) that the contribution from such quantities can be small, one cannot generally rule out possibilities of larger value contributions stemming from $||\frac{\partial U(\vartheta)}{\partial \vartheta}||_\Diamond$ as stated in the preceding rebuttals.
> > >
> > > Moreover, as pointed out by the referee, it sounds very reasonable to take into account the "curvature" effects of the model space/ or Hessian terms that are encapsulated in the $\mathcal{O}(\delta^2)$ (Although as explained above, it is not necessarily because the norms are minimized, but rather captures geometric aspects of the underlying model space), it is beyond the scope of this paper, since we put-forth as a *first step a new figure of merit to analyze the model space in terms of channel distinguishability*. Nonetheless, our proposed metric is *not* “necessarily” locally flat in the neighborhood of zero $|\delta - 0| < \epsilon$, i.e.,
> > >
> > > $(\nabla_{\vartheta}{\|A(\vartheta)\|_\Diamond}) \ne 0$
> > >
> > > (evaluated at $\delta = 0$) but we considered only the first order contributions (Jacobian/gradients) akin to perturbative expansions, mainly due to (a) insufficient existing theory on the Hessians and higher-order derivatives associated with diamond norms predominantly linked to "curvature" and higher-order effects induced corrections to the channel sensitivity, and (b) numerical instabilities in computational algorithms for computing the same. Thus, it is again not necessary to go to at least to the $\mathcal{O}(\delta^2)$ for finding non-vanishing/finite contributions.
> > >
> > > Also, the analogy to the classical ML case, wherein the derivatives of the KL-divergence between probabilistic measures under parameter perturbations requires the Fisher-Information Matrix coming from the second order corrections to obtain meaningful non-zero terms is promising, the KL divergence is an easier quantity, and it is straightforward to show that under parameter updates the first derivatives identically vanish. On the contrary, the diamond norms depend on the eigenspectrum with derivatives not always vanishing/well-defined.
> > >
> > > This is indeed a good suggestion by the referee, and the sensitivity associated to Fisher information (Hessian) contributions in the $\mathcal{O}(\delta^2)$ expansion will be taken into account and integrated in future works.
> > >
> > > Thus, with the above arguments we intend to differ from the view-point that was suggested (although, we acknowledge that it is an alternative way to proceed) and put-forth that this is not a mere mathematical artifact of the norms being small, i.e. $\lim_{\delta \rightarrow 0}||\delta|| = 0$.  Subsequently, the gradient $||(\nabla_\delta(||f(\theta) - f(\theta + \delta))||))|_{\delta=0} \neq 0|| \ \forall f \in C^{\infty}(\mathbb{R}^d)$ does not identically vanish for all cases of small perturbations and for different architectures need not be a *flat* quantity.
> > >
> > > This signifies that it is an "innate" property of the QNN architectures or the ansatze the end-user chooses as their quantum models since the corresponding derivatives on the spectral manifolds (eigenspectrum or singular values or eigenvalue) $\{ \sigma_i(A) \}_{1 \leq i \leq n}$ appearing in the diamond norm evaluated at or near critical points can be significantly influenced by the ansatz itself. Moreover, the $\mathcal{O}(\delta^2)$ is can give rise to significant contributions (or even higher order terms in the series expansion), which will be considered in detail in future works.
> > >
> > > We hope that we have clarified the misunderstandings now. Again, we highly appreciate the active discussion with the reviewer and want to thank them for their time and thorough comments.

---

> ### Comment · Reviewer_U5bz · 2024-11-28
>
> Thanks to the authors for a convincing reply. I appreciate the authors constructive engagement.
>
> Indeed it is likely the case that the diamond norm (and other norms, as pointed out by the authors) is not analytic around zero, consequently cannot be expanded around zero as I had done in my previous comments. The authors rebuttal on this is convincing and addresses my principal concern.
>
> My only remaining hesitations are not to do with correctness, and can be addressed with rewordings/discussion. I apologise to the authors for not mentioning these earlier; they are thoughts that have arisen during the discussion phase as I have understood the work better.
>
> In short, I find the channel distinguishability and its bounds to be interesting and relevant. I also think the experiments are reasonably done given the difficulty of computing diamond norms. But I feel that some care needs to be taken in drawing the right conclusions which can be addressed with minor rewordings and some discussion points:
>
> - On contribution 3 (line 69 in the updated version). The claim is phrased as pertaining to HEA in general without mention of the empirical setting of the numerical experiments. The numerical evidence only applies to circuit sizes of 1-4 qubits which are very small for simulations on even consumer level hardware---NISQ QML practitioners should at least expect classical intractability of a state vector simulation before expecting any chance of quantum advantage or utility, so maybe 50-100 qubits? This is well beyond what is (or can be) considered in this work. Consequently, it feels unfair to apply the empirical results as general criticism of HEA on NISQ circuits in general. The scale and context of the experiments need to be mentioned in the contribution 3 to avoid misinterpretation.
> - Related to above: some discussion the interplay between diamond norms and circuit sizes would be beneficial. It feels natural that for small circuit sizes diamond norms should be smaller irrespective of ansatz: the diamond norm is a measure of single-use distinguishability, and there is little information to be acquired from a single measurement of a small circuit. Does this induce a system-size related dependence on the channel distinguishability? Is the looseness of the paper's upper bounds when compared to experiments perhaps tied to small circuits considered empirically?
> - What level of channel distinguishability should be expected on parameter update? What is healthy and what is not? To my knowledge, this is the first study considering the channel distinguishability on parameter update for variational quantum algorithms. How much distinguishability should there be from a "good" ansatz that is learning well? The authors provide interesting arguments in earlier replies that there should be ansätze with higher channel distinguishability than HEA, but what do the channel distinguishabilities of these ones actually look like empirically under parameter update? As we've seen in the paper, there might be little correspondence between the first-order bounds and empirical praxis. Before concluding that HEA channel distinguishability is small, it needs to be established what counts as small in the first place. Experiments along these lines are beyond the scope of this paper, but I think this point should be discussed.
> - Contribution 4 also needs more care in construction. I do not see the similarity with barren plateaus. BPs directly consider the gradients of parameters with respect to a loss function. But this work eschews a loss function altogether. In the absence of understanding what a healthy channel distinguishability should be for a parameter update (see above point), and considering the limited nature of the experiments, how can any general conclusions on trainability be made?
> - Including the numerical values for the upper bounds for each ansatz within the figure captions would be really helpful for readers.
>
> Thanks again to the authors for their constructive engagement.

---

> > ### Author Response · Authors · 2024-11-29
> > **Part 1**
> >
> > We thank the referee for their answer and are happy to have clarified our work. We are glad to address the open comments as outlined below, however, considering that the rebuttal period has passed, we could only do that in the camera ready version, if the paper is accepted. We hope this is a satisfactory solution for the referee, and that they consider raising their rating in the meantime.
> >
> > * Ad 1). We agree with the referee that we need to be more specific in this point and acknowledge the important point raised. We are happy to change the phrasing to the following, noting that we would keep the discussion based on the number of parameters (which implicitly includes both system size and depth) and is what we base both theory and experiments on. As an example, channel sensitivity might also be large on a very deep 4 qubit setting, hence, arguing based on only system size might not be fully accurate. "We provide numerical evidence that contemporary QNNs with HEA with a sufficiently modest number of parameters are hardly distinguishable upon parameter update in (1) random perturbations and (2) during training, even in early stages, with larger updates".
> > * Ad 2). While we agree with the referee that this would be very helpful, we are unable to provide such a discussion. The diamond norm is a highly nontrivial measure, and such results are not yet available. As we showed in our work, the channel sensitivity can be related to the number of parameters in the circuit, which is already an interesting insight and is, to the best of our knowledge, novel. Beyond that, diamond norms between slightly perturbed channels (or even from channels generated by the same function) have not been studied. There is a body of knowledge on diamond norms between random channels (e.g., arXiv:1612.00401), which, under probability distribution assumptions, grows in the system size. However, such use cases are not related to our work (as the assumptions are definitely not met), and it would be misleading to reference them. The diamond norm does, however, not generally grow in system size, as an example, the diamond norm between a 1-qubit X gate and Z gate is also 2. We acknowledge that this is an incredibly interesting research direction that we would like to consider in the future, but there is no sufficient knowledge on the topic now to be able to discuss this, even from a numerical standpoint. We are happy to add a paragraph discussing what we outlined above to the paper.
> > * Ad 2b) To touch on the single use distinguishability, we would like to recap our previous response: .. the diamond norm is defined according to the single use distinguishability, but there are implications for the expectation value (ensemble average) based on the single-use case (if we have a 5% chance of distinguishing them in a single use, this has implications for how often the result will be different when averaging over 100s of shots)

---

> > > ### Author Response · Authors · 2024-11-29
> > > **Part 2**
> > >
> > > * Ad 3). This is an excellent point that we could definitely touch on. While we cannot put an exact number on the channel sensitivity for a "good" ansatz, we can definitely discuss what should be expected. In particular, intuitively, an ansatz should be distinguishable from itself upon update, as, for trainability, we should at least be able to see a difference in results. This touches upon the reviewer's fourth point; the resemblance to BPs we mention is exactly due to this behavior. If the upper bound we can find (which seems to not be achieved in large parts of the model space, as observed in the experiments) is low already, i.e., the upper bound on the maximum distinguishability already shows that it is very unlikely to observe a difference on *any* input state, we can expect for the expectation value of our circuit to not (or hardly) to change. If this is the case in the local neighborhood of the current step, we are plateauing, which has direct implications on the loss landscape (due to the composition of mappings, cf. Fig 2. in arXiv:2109.11676). We formalize this further in the next bullet point. Further, we want to mention that this as well is an interesting research area, as this seems to be an interesting interplay between being able to distinguish subsequent quantum channels and the value of the loss function still changing only slowly to ensure efficient training.
> > > * Ad 4). We agree with the reviewer that we were not careful enough with this statement. As discussed above, the resemblance exists (i.e., if the unitaries in the local neighborhood lead to undistinguishable results on *any* input state, we are inevitably plateauing; this is independent of an exact value of a "healthy" distinguishability), however, the point requires more careful wording and more elaboration. Upon acceptance, we would adapt both the contributions part and the discussion section in that regard in more formal way as follows.
> > >     "Adapting a series of composition of maps viewpoint (arXiv:2109.11676), our work argues that the unitaries in the unitary space of the considered architectures do not change significantly upon parameter update in the local neighborhood of the parameter space. Thus, from small channel distinguishability, it follows that the quantum states in the Hilbert space will not differ significantly, as will the expectation values of the circuit (which is a consequence of the diamond norm formulation), which are directly related to the loss. Roughly speaking, this behavior is mirrored in the Barren Plateau phenomenon. While we do not argue for strict equivalence, the similarities hint towards two perspectives on similar trainability issues. Therefore, this phenomenon might not only be tied to the exponential Hilbert space, but may be an artifact of the architectures themselves."
> > > * Regarding the last comment, as per request of Reviewer kPLW, we moved this figure to the appendix and replaced it with one without the upper bound. Still, the figure with the upper bound can be seen in Appendix D. We are happy to add a more explicit reference to this figure in the main text.

---

> ### Comment · Reviewer_U5bz · 2024-11-29
>
> Thank you authors for the continued dialogue throughout the extended discussion period.
>
> I trust the remaining rewordings/discussion will be incorporated for a camera ready version.
>
> The diamond norm is an interesting and relevant way of scrutinising most any QML model. Much remains to be done to see its full relevance to QML in general. I hope this will be the first of more papers to come on the subject. I have updated my recommendation accordingly to a clear acceptance.

---

### Official Review · Reviewer_BvBg · 2024-11-02

**Soundness:** 3
**Presentation:** 3
**Contribution:** 3
**Rating:** 8
**Confidence:** 4

**Summary:**

The study presents the authors' work on evaluating the expressivity of different ansatzes, while trying to propose novel measures and metrics to capture the behaviour of QNNs during training.

**Strengths:**

The paper looks from a different angle to the well-known problem of training QNNs and presents new approaches to assessment of quantum model expressivity while discussing the need to develop architectures suitable for QML. This seems like an important work for the QML community.

**Weaknesses:**

The paper did not explain what particular elements or spatial organisation of gates of ansatzes may be the most contributing to expressivity or trainability issues observed in QNNs.

**Questions:**

Do the authors envision that researchers in the field would be able to leverage their findings to build better application-specific models? Could standard architecture search approaches be coupled with suitable distinguishability metrics to build better ansatzes?

What proportion of the expressivity issues observed with QML models are related to the ansatz vs the data encoding?

As all experiments seem to be classical simulations, do the authors have an idea how much will noise when executing the model on a device contribute to expressivity issues?

---

> ### Author Response · Authors · 2024-11-20
>
> Initially, we would like to thank the reviewer for their time and valuable comments. We acknowledge that the (ansatz) expressivity concept is not 100\% clearly stated in the work, which we will therefore add it as follows. We hope this clarifies some aspect of the work better.
>
> Ansatz Expressivity: "We acknowledge a missing clear definition of expressivity in QML for now. It is sensible, therefore, to transfer it from classical ML, where model expressivity is defined as the range of functions a model is able to compute (arXiv:1606.05336). Hence, it is a function of architecture $A$, input $x$ and parameters ${\vartheta}$; $F_A(x;{\vartheta})$. On the contrary, our work lies in the premise of data-agnostic characterization of models. To this end, we will only consider *ansatz expressivity*; the set of functions that can be generated solely by an ansatz itself."
>
> Issues with definition: "If one were to define expressivity as the capability of a model to represent all possible set of unitaries, it would be required to go way beyond second moments (for, e.g., Kurtosis, off-diagonal correlations, etc. ) to ensure the necessary weights are captured where there is sufficiently dense population. As the moment operator requires tensor products proportional to the moments, this becomes computationally infeasible very quickly. It is not clear how to overcome the curse of dimensionality while at the same time proposing a measure that captures expressivity adequately; in fact, most tools from QIT suffer from this very phenomenon, which speaks to the difficulty of designing such a metric. Further, there already exists a line of non-unitary VQA research (e.g., arXiv:1810.03787, arXiv:2411.05760), that needs to be considered as well.
>
> While particular elements and spatial organisation of gates and their influence on the model are indeed an interesting area of research, it was not the focus of our work to consider expressivity (due to missing criteria and even definition) or trainability issues for specific architectures (state-of-the-art attributes these to the high dimensional space (=concentration of measure), and if they are trainable, they seem to encode it into a polynomial subspace (=classically simulable); arXiv:2312.09121).
>
> Questions:
>
> 1. The main contribution of our work is showing that small HEAs, as they are used today, are difficult to train from an architectural point of view (without data), due to the unitaries hardly changing upon parameter update. The criterion we came up with can absolutely be used to evaluate future VQA architectures; considering we do not use input data at all, not necessarily application-specific but rather general classes of architectures. With that in mind, it could surely also be used for architecture search (considering computational complexity though).
>
> 2. We do not understand the term "expressivity issues", as the main issue we pointed out was that the currently used measure fails at capturing this very property; but there are no issues with expressivity generally. Therefore, we would like to ask the reviewer to specify the term "expressivity issues" more concretely, if the following does not answer the question. Generally, from the classical ML community (cf. arXiv:1606.05336), it refers to the range of functions a model is able to express; therefore, is dependent on both, i.e., the architecture/model as well as the input data. As we will state more clearly in the rebuttal version (see above), we refer to ansatz expressivity in this work.
>
> 3. Expressivity of a model is a purely theoretical concept that is independent of errors occurring during training runs. Therefore, the expressivity is not affected by the noise (again, we are not sure what the issues w.r.t expressivity refers to; we would be grateful for a clarification of that term). Generally, the observed channel sensitivity in training runs will be affected by the noise as the expectation values are affected, hence, so will the parameter updates; this does not invalidate our bounds or work generally, though.

---

> > ### Comment · Reviewer_BvBg · 2024-11-26
> > **Terms clarified**
> >
> > I would like to thank the authors for the detailed explanations and clarification of certain terms.

---

### Author Response · Authors · 2024-11-27
**Rebuttal Letter**

We would like to, once more, thank the reviewers for their insightful advice and comments. We have uploaded a new version of the paper, highlighting the changes in yellow.

In particular:
* We added more elaboration on the concept of expressivity and our usage thereof. We also more explicitly stated that we consider ansatz expressivity in the paper.
* We clarified the terminology in Section 3 and added clearer interpretation and discussion on the concept of expressivity.
* We rearranged the derived bound (previously in Section 6) into Section 4 for a better structure.
* Information for the experiments (optimizer + hyperparameters), as well as for the confidence intervals, was added.
* We redrew Figure 3 without the bound. Another appendix was added to include the previous figure to show the discrepancy between observed channel sensitivity and the bound. We clarified that this does indeed not mean that the bound is loose, as it may be achieved in some parts of the model space. Further, we replaced the figures with PDF version for better readability.
* The notational inconsistencies in Eqs. 25 and 26 were fixed as requested by Reviewer kPLW.
* Further, we include the results of the experiments with angle encoding in the appendix (as requested by Reviewers kPLW and U5bz). We choose to keep them in the appendix, as they do not lead to significant new findings. Rather, it showcases robustness of our experiments w.r.t. feature encoding.

We will address the reviewers individually in the comments sections for open questions.

---

### Meta-Review · Area_Chair_faxj · 2024-12-19

**Metareview:**

The paper tackles a critical question in quantum neural networks (QNNs), specifically the evaluation of ansatz properties and the challenges associated with training. To this end, the authors introduced the concept of channel sensitivity as a metric for assessing ansatz characteristics. The results present a novel and theoretically robust contribution to the field. While limitations in scalability and practical applicability were acknowledged, the authors' responses successfully addressed most concerns, particularly those related to clarity and experimental rigor. Overall, the paper makes a valuable contribution to the quantum machine learning community.

**Additional Comments On Reviewer Discussion:**

The authors and reviewers engaged in discussions addressing several critical points. These included the lack of clarity in defining expressivity and its relation to the proposed channel sensitivity metric, the scalability challenges associated with using the diamond norm as a practical tool, and the limited experimental scope, which primarily focused on small HEA architectures without exploring larger or alternative ansatz designs. Following the rebuttal period, the authors and reviewers largely reached a consensus on these issues.

---

### Decision · Program_Chairs · 2025-01-22

Accept (Poster)